# Interpretable Graph Networks Formulate Universal Algebra Conjectures

**Francesco Giannini**\*
CINI, Italy
francesco.giannini@unisi.it

**Stefano Fioravanti**\*
Università di Siena, Italy, JKU Linz, Austria Italy
stefano.fioravanti@unisi.it

**Oguzhan Keskin**
University of Cambridge, UK
ok313@cam.ac.uk

**Alisia Maria Lupidi**
University of Cambridge, UK
aml201@cam.ac.uk

**Lucie Charlotte Magister**
University of Cambridge, UK
lcm67@cam.ac.uk

**Pietro Lió**
University of Cambridge, UK
pl219@cam.ac.uk

**Pietro Barbiero**\*
Università della Svizzera Italiana, CH
University of Cambridge, UK
barbip@usi.ch

## Abstract

The rise of Artificial Intelligence (AI) recently empowered researchers to investigate hard mathematical problems which eluded traditional approaches for decades. Yet, the use of AI in Universal Algebra (UA)—one of the fields laying the foundations of modern mathematics—is still completely unexplored. This work proposes the first use of AI to investigate UA's conjectures with an equivalent equational and topological characterization. While topological representations would enable the analysis of such properties using graph neural networks, the limited transparency and brittle explainability of these models hinder their straightforward use to empirically validate existing conjectures or to formulate new ones. To bridge these gaps, we propose a general algorithm generating AI-ready datasets based on UA's conjectures, and introduce a novel neural layer to build fully interpretable graph networks. The results of our experiments demonstrate that interpretable graph networks: (i) enhance interpretability without sacrificing task accuracy, (ii) strongly generalize when predicting universal algebra's properties, (iii) generate simple explanations that empirically validate existing conjectures, and (iv) identify subgraphs suggesting the formulation of novel conjectures.

## 1 Introduction

Universal Algebra (UA, (6)) is one of the foundational fields of modern Mathematics with possible deep impact in all mathematical disciplines, but the complexity of studying abstract algebraic structures hinders scientific progress and discourages many academics. Recently, the emergence of powerful AI technologies empowered researchers to investigate intricate mathematical problems which eluded traditional approaches for decades, leading to the solution of open problems (e.g., (24)) and discovery of new conjectures (e.g., (7)). Yet, universal algebra currently remains an uninvestigated realm for AI, primarily for two reasons (i) first

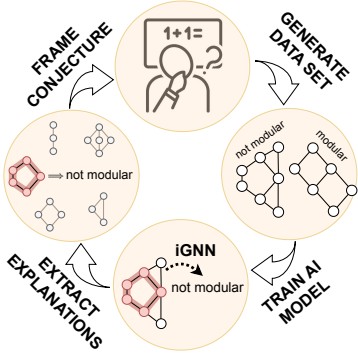

Figure 1: Interpretable graph networks support universal algebra research.

37th Conference on Neural Information Processing Systems (NeurIPS 2023).

UA deals with infinite objects or even classes of abstract objects, which pose unique challenges for conventional AI techniques, (ii) secondly the field commonly relies on deterministic algorithms, utilized to construct finite models or serve as deterministic theorem provers, such as "mace4" and "prover9"[1]. In this sense, we hope that our paper could represent a guidance to further explore the study of UA's problems with AI, e.g. by integrating existing systems, like these mentioned theorem provers, with our methodology, which may suggest novel conjectures, in a unique scheme.

Universal algebra studies algebraic structures from an abstract perspective. Interestingly, several UA conjectures equivalently characterize algebraic properties using equations or graphs (16). In theory, studying UA properties as graphs would enable the use of powerful AI techniques, such as Graph Neural Networks (GNN, (39)), which excel on graph-structured data. However, two factors currently limit scientific progress. First, the *absence of benchmark datasets* suitable for machine learning prevents widespread application of AI to UA. Second, *GNNs' opaque reasoning* obstructs human understanding of their decision process (38). Compounding the issue of GNNs' limited transparency, GNN explainability methods mostly rely on brittle and untrustworthy local/post-hoc methods (14; 27; 28; 38; 45) or pre-defined subgraphs for explanations (2; 42), which are often unknown in UA.

**Contributions.** In this work, we investigate universal algebra's conjectures through AI (Figure 1). Our work includes three significant contributions. First, we propose a novel algorithm that generates a dataset suitable for training AI models based on an UA equational conjecture. Second, we generate and release the first-ever universal algebra's dataset compatible with AI, which contains more than $29,000$ lattices and the labels of $5$ key properties i.e., modularity, distributivity, semi-distributivity, join semi-distributivity, and meet semi-distributivity. And third, we introduce a novel neural layer that makes GNNs fully interpretable, according to Rudin's (38) notion of interpretability. The results of our experiments demonstrate that interpretable GNNs (iGNNs): (i) enhance GNN interpretability without sacrificing task accuracy, (ii) strongly generalize when trained to predict universal algebra's properties, (iii) generate simple concept-based explanations that empirically validate existing conjectures, and (iv) identify subgraphs which could be relevant for the formulation of novel conjectures. As a consequence, our findings demonstrate the potentiality of AI methods for investigating UA problems.

## 2 Background

Universal Algebra is a branch of mathematics studying general and abstract algebraic structures. *Algebraic structures* are typically represented as ordered pairs $\mathbf{A} = (A, F)$, consisting of a non-empty set $A$ and a collection of operations $F$ defined on the set. UA aims to identify algebraic properties (often in equational form) shared by various mathematical systems. In particular, *varieties* are classes of algebraic structures sharing a common set of identities, which enable the study of algebraic systems based on their common properties. Prominent instances of varieties that have been extensively studied across various academic fields encompass Groups, Rings, Boolean Algebras, Fields, and many others. A particularly relevant variety of algebras are Lattices (details in Appendix A.3), which are often studied for their connection with logical structures.

**Definition 2.1.** A *lattice* $\mathbf{L}$ is an algebraic structure composed by a non-empty set $L$ and two binary operations $\vee$ and $\wedge$, satisfying the commutativity, associativity, idempotency, and absorption axioms.

Equivalently a lattice can be characterized as a partially ordered set in which every pair of elements has a *supremum* ($\vee$) and an *infimum* ($\wedge$) (cf. Appendix A.3). Lattices also have formal representations as graphs via *Hasse diagrams* $(L, E)$ (e.g., Figure 2), where each node $x \in L$ is a lattice element, and directed[2] edges $(x, y) \in E \subseteq L \times L$ represent the ordering relation, such that if $(x, y) \in E$ then $x \leq_L y$ in the ordering of the lattice. A *sublattice* $\mathbf{L}'$ of a lattice $\mathbf{L}$ is a lattice such that $L' \subseteq L$ and

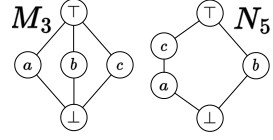

Figure 2: Hasse diagrams.

$\mathbf{L}'$ preserves the original order (the "essential structure") of $L$, i.e. for all $x, y \in L'$ then $x \leq_{L'} y$ if and only if $x \leq_L y$. The foundational work by Birkhoff (5), Dedekind (9), and Jónsson (18) played a significant role in discovering that some significant varieties of lattices can be characterized through the omission of one or more lattices. Specifically, a variety $\mathcal{V}$ of lattices is said to *omit* a lattice $\mathbf{L}$ if the latter cannot be identified as a sublattice of any lattice in $\mathcal{V}$. A parallel line of work in UA

---

[1] https://www.cs.unm.edu/~mccune/prover9/.

[2] The orientation of Hasse diagrams is always to be meant bottom-up, hence we will omit arrows for simplicity.

characterizes lattices in terms of equational ("$term_1 \approx term_2$") and quasi-equational ("if $equation_1$ holds then $equation_2$ holds") properties, such as distributivity and modularity.

**Definition 2.2.** Let **L** be a lattice. **L** is *modular* if it satisfies $x \leq y \rightarrow x \vee (y \wedge z) \approx y \wedge (x \vee z)$; *distributive* if it satisfies $x \vee (y \wedge z) \approx (x \vee y) \wedge (x \vee z)$.

We notice that each distributive lattice is also modular (see Appendix A.3). Figure 2 represents $\mathbf{M}_3$ (modular with 3 atoms) and $\mathbf{N}_5$ (non-modular with 5 atoms), the smallest instances of lattices which exhibit failures of distributivity and modularity, respectively. For example, $\mathbf{N}_5$ is neither modular nor distributive, considering the substitution $x = a, y = c, z = b$. The same substitution shows that $\mathbf{M}_3$ is not distributive. The classes of distributive and modular lattices show classical examples of varieties that can equivalently be characterized using equations and lattice omissions, as illustrated by the following theorems.

**Theorem 2.3** (Dedekind (9)). *A lattice variety $\mathcal{V}$ is modular if and only if $\mathcal{V}$ omits $\mathbf{N}_5$.*

**Theorem 2.4** (Birkhoff (5)). *A lattice variety $\mathcal{V}$ is distributive if and only if $\mathcal{V}$ omits $\mathbf{N}_5$ and $\mathbf{M}_3$.*

Starting from these classic results, the investigation of lattice omissions and the structural characterizations of classes of lattices has evolved into a rich and extensively studied field (16), but it was never approached with AI methodologies before.

# 3 Methods

The problem of characterizing lattice varieties through lattice omission is very challenging as it requires the analysis of large (potentially infinite) lattices (5; 9; 18). To address this task, we propose the first AI-assisted framework supporting mathematicians in finding empirical evidences to validate existing conjectures and to suggest novel theorems. To this end, we propose a general algorithm (Section 3.1) allowing researchers in universal algebra to define a property of interest and generate a dataset suitable to train AI models. We then introduce interpretable graph networks (Section 3.2) which can suggest candidate lattices whose omission is responsible for the satisfaction of the given algebraic property.

## 3.1 A Tool to Generate Datasets of Lattice Varieties

We propose a general methodology to investigate any algebraic property whose validity can be verified on a finite lattice. In this work, we focus on properties that can be characterized via equations and quasi-equations. To train AI models, we propose a general dataset generator[3] for lattice varieties. Intuitively, the generator takes as input the number of nodes $n$ in the lattices and a function to check whether a lattice satisfies a given property. We generate $2^{n \times n}$ matrices of size $n \times n$, containing all binary functions

---

**Algorithm 1:** Generate dataset of lattice varieties.

**Input:** $n \geq 1$, $hasProperty(\cdot, \cdot, \cdot)$       // $n$: cardinality
$Dataset = []$
$AllFuncs \leftarrow genAllFuncs(n)$    // binary functions as $n \times n$ matrices
**for** $L \in AllFuncs$ **do**      // $L(i, j) = 1$ meaning $i \leq_L j$
  **if** $isPartialOrder(L)$ **then**    // check if $\leq_L$ is refl., antisym. and trans.
    **if** $isLattice(L)$ **then**      // check if $L$ is a lattice
      **for** $i, j \leq n$ **do**
        $\wedge_L[i, j] \leftarrow \sup_{x \leq_L} \{x \leq_L i \text{ and } x \leq_L j\}$
        $\vee_L[i, j] \leftarrow \inf_{x \leq_L} \{i \leq_L x \text{ and } j \leq_L x\}$
      **if** $hasProperty(L, \wedge_L, \vee_L)$ **then**   // check $\wedge_L, \vee_L$ properties
        $Dataset.append([L, \text{True}])$
      **else**
        $Dataset.append([L, \text{False}])$

---

definable on $\{1, \ldots, n\}^2$, and filter only binary matrices representing partial orders. Then, we verify that the partial ordered set $L$ is a lattice, by checking that any pair of nodes always has a unique infimum and supremum. This directly verifies that $\wedge_L$ and $\vee_L$ satisfy Definition 2.1. Finally, we check whether the lattice satisfies the target property or not, and append it and the property label to our dataset. We remark that checking the validity of a single ternary equation on a medium-size lattice is not computationally prohibitive (i.e., it "only" requires checking $n^3$ identities), but the number of existing lattices increases exponentially as $n$ increases. For instance, it is known that there are at least 2,000,000 non-isomorphic lattices with $n = 10$ elements (4). Therefore, we only sample a fixed number of lattices per cardinality starting from a certain node cardinality. While this may seem a strong bias, we notice that known and relevant lattice omissions often rely on lattices with few nodes (5; 9). To empirically verify that this is not a significant limitation, in our experiments we deliberately investigate the generalization capacity of GNNs when trained on small-size lattices and tested on larger ones. This way we can use GNNs to predict the satisfiability of equational properties

---

[3]The dataset generator and the datasets are available at `https://github.com/fragiannini/AI4UA`.

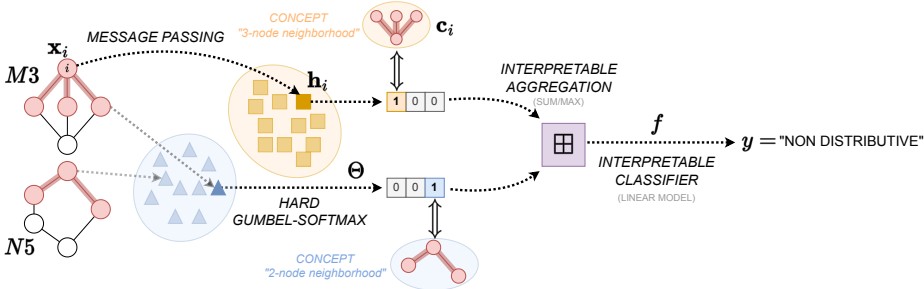

Figure 3: An interpretable graph layer (i) aggregates node features with message passing, (ii) generates a node-level concept space with a hard Gumbel-Softmax activation $\Theta$, (iii) generates a graph-level concept space with an interpretable permutation invariant pooling function $\boxplus$ on node-level concepts, and (iv) predicts a task label with an interpretable classifier $f$ using graph-level concepts.

on large graph structures without explicitly checking them. Using Algorithm 1, we generated the first large-scale AI-compatible datasets of lattices containing more than $29,000$ graphs and the labels of 5 key properties of lattice (quasi-)varieties i.e., modularity, distributivity, semi-distributivity, join semi-distributivity, and meet semi-distributivity, whose definitions can be found in Appendix A.

**Scalability and Complexity.** The scalability of the dataset generation process is related to two main points: (i) dealing with the exponential growth of the number of lattices, (ii) how to feasibly generate a lattice of a finite amount of elements. (i) Since the number of lattices definable on a set $L = \{0, ...n - 1\}$ increases exponentially with $n$, it is not feasible to realize a dataset containing all the lattices of $n$ elements. However, this is also unnecessary for the scope of this work, which aims at automatically identifying (small) topological patterns responsible for the failure of certain algebraic properties. In addition, the majority of the graphs generated are isomorphic, thus not particularly informative for our task. Because of this, we only generate a portion of all the lattices up to a certain value $M > 0$, and then take a fixed number of samples $n_s$ up to $N$ (we take $M = 8$, $n_s = 20$, $N = 50$ in the paper). (ii) The construction of a lattice with $n$ elements is briefly sketched in Algorithm 1. We refer to Appendix B for the technical details on the design of the used functions.

### 3.2 Interpretable Graph Networks (iGNNs)

In this section, we design an interpretable graph network (iGNN, Figure 3) that satisfies the notion of "interpretability" introduced by Rudin (38). According to this definition, a machine learning (ML) system is interpretable if and only if (1) its inputs are semantically meaningful, and (2) its model inference is simple for humans to understand (e.g., sparse and/or symbolic). This definition covers ML systems that take tabular datasets or sets of concepts as inputs, and (piece-wise) linear models such as logistic regression or decision trees. To achieve this goal in GNNs, we introduce an interpretable graph layer that learns semantically meaningful concepts and uses them as inputs for a simple linear classification layer. We then show how this layer can be included into existing architectures or into hierarchical iGNNs, which consist of a sequence of interpretable graph layers. We notice that in devising our approach, we preferred to rely on linear classifiers as they are fully differentiable, hence allowing us to realize a fully interpretable and differentiable model from the input to the classification head. However,in practice any interpretable and differentiable classifier could be used in place of this linear layer.

#### 3.2.1 Interpretable Graph Layer

The interpretable graph layer (Figure 3) serves three main functions: message passing, concept generation, and task predictions. The first step of the interpretable graph layer involves a standard message passing operation (Eq. 1 right), which aggregates information from node neighbors. This operation enables to share and process relational information across nodes and it represents the basis of any GNN layer.

**Node-level concepts.** An interpretable concept space is the first step towards interpretability. Following Ghorbani et al. (10), a relevant concept is a "high-level human-understandable unit of

information" shared by input samples and thus identifiable with clustering techniques. Message passing algorithms do cluster node embeddings based on the structure of node neighborhoods, as observed by Magister et al. (28). However, the real-valued large embedding representations $\mathbf{h}_i \in \mathbb{R}^q, q \in \mathbb{N}$ generated by message passing can be challenging for humans to interpret. To address this, we use a hard Gumbel-Softmax activation $\Theta : \mathbb{R}^q \mapsto \{0, 1\}^q$, following Azzolin et al. (2):

$$\mathbf{c}_i = \Theta(\mathbf{h}_i) \qquad \mathbf{h}_i = \phi\Big(\mathbf{x}_i, \bigoplus_{j \in N_i} \psi(\mathbf{x}_i, \mathbf{x}_j)\Big) \tag{1}$$

where $\psi$ is a learnable function ignoring or assuming constant input features and $\phi$ is a learnable function aggregating information from a node neighborhood $N_i$, and $\oplus$ is a permutation invariant aggregation function (such as sum or mean). During the forward pass, the Gumbel-Softmax activation $\Theta$ produces a one-hot encoded representation of each node embedding. Since nodes sharing the same neighborhood have similar embeddings $\mathbf{h}_i$ due to message passing, they will also have the same one-hot vector $\mathbf{c}_i$ due to the Gumbel-Softmax, and vice versa - we can then interpret nodes having the same one-hot concept $\mathbf{c}_i$ as nodes having similar embeddings $\mathbf{h}_i$ and thus sharing a similar neighborhood. More formally, we can assign a semantic meaning to a reference concept $\gamma \in \{0, 1\}^q$ by visualizing concept prototypes corresponding to the inverse images of a node concept vector. In practice, we can consider a subset of the input lattices $\Gamma$ corresponding to the node's ($p$-hop) neighborhood covered by message passing:

$$\Gamma(\gamma, p) = \Big\{ \mathbf{L}^{\langle i,p \rangle} \mid i \in L \land \mathbf{L} \in \mathcal{D} \land \mathbf{c}_i = \gamma \Big\} \tag{2}$$

where $\mathcal{D}$ is the set of all training lattices, and $\mathbf{L}^{\langle i,p \rangle}$ is the graph corresponding to the $p$-hop neighborhood ($p \in \{1, \ldots, |L|\}$) of the node $i \in L$, as suggested by Ghorbani et al. (10); Magister et al. (28). This way, by visualizing concept prototypes as subgraph neighborhoods, the meaning of the concept representation becomes easily interpretable to humans (Figure 3), aiding in the understanding of the reasoning process of the network.

**Example 3.1** (Interpreting node-level concepts). Consider the problem of classifying distributive lattices with a simplified dataset including $\mathbf{N}_5$ ⬡ and $\mathbf{M}_3$ ⬡ only, and where each node has a constant feature $x_i = 1$. As these two lattices only have nodes with 2 or 3 neighbours, one layer of message passing will then generate only two types of node embeddings e.g., $\mathbf{h}_{II} = [0.2, -0.4, 0.3]$ for nodes with a 2-nodes neighborhood (e.g., ⬡), and $\mathbf{h}_{III} = [0.6, 0.2, -0.1]$ for nodes with a 3-nodes neighborhood (e.g., ⬡). As a consequence, the Gumbel-Softmax will only generate two possible concept vectors e.g., $\mathbf{c}_{II} = [0, 0, 1]$ and $\mathbf{c}_{III} = [1, 0, 0]$. Hence, for instance the concept ⬡ belongs to $\mathbf{c}_{II}$, while ⬡ belongs to $\mathbf{c}_{III}$.

**Graph-level concept embeddings.** To generate a graph-level concept space in the interpretable graph layer, we can utilize the node-level concept space produced by the Gumbel-Softmax. Normally, graph-level embeddings are generated by applying a permutation invariant aggregation function on node embeddings. However, in iGNNs we restrict the options to (piece-wise) linear permutation invariant functions in order to follow our interpretability requirements dictated by Rudin (38). This restriction still includes common options such as max or sum pooling. Max pooling can easily be interpreted by taking the component-wise max over the one-hot encoded concept vectors $\mathbf{c}_i$. After max pooling, the graph-level concept vector has a value of $1$ at the $k$-th index if and only if at least one node activates the $k$-th concept i.e., $\exists i \in L, \mathbf{c}_{ik} = 1$. Similarly, we can interpret the output of a sum pooling: a graph-level concept vector takes a value $v \in \mathbb{N}$ at the $k$-th index after sum pooling if and only if there are exactly $v$ nodes activating the $k$-th concept i.e., $\exists i_0, \ldots, i_v \in L, \mathbf{c}_{ik} = 1$.

**Example 3.2** (Interpreting graph-level concepts). Following Example 3.1, let us use sum pooling to generate graph-level concepts. For an $\mathbf{N}_5$ graph, we have 5 nodes with exactly the same 2-node neighborhood. Therefore, sum pooling generates a graph-level embedding $[0, 0, 5]$, which certifies that we have 5 nodes of the same type e.g., ⬡. For an $\mathbf{M}_3$ graph, the top and bottom nodes have a 3-node neighborhood e.g., ⬡, while the middle nodes have a 2-node neighborhood e.g., ⬡. This means that sum pooling generates a graph-level embedding $[2, 0, 3]$, certifying that we have 2 nodes of type ⬡ and 3 nodes of type ⬡.

**Interpretable classifier.** To prioritize the identification of relevant concepts, we use a classifier to predict the task labels using the concept representations. A black-box classifier like a multi-layer perceptron (36) would not be ideal as it could compromise the interpretability of our model, so instead we use an interpretable linear classifier such as a single-layer network (20). This allows for a completely interpretable and differentiable model from the input to the classification head, as the input representations of the classifier are interpretable concepts and the classifier is a simple linear model which is intrinsically interpretable as discussed by Rudin (38). In fact, the weights of the perceptron can be used to identify which concepts are most relevant for the classification task. Hence, the resulting model can be used not only for classification, but also to interpret and understand the problem at hand.

### 3.2.2 Interpretable architectures

The interpretable graph layer can be used to instantiate different types of iGNNs. One approach is to plug this layer as the last message passing layer of a standard GNN architecture:

$$\hat{y} = f\left( \boxplus_{i \in K} \ \left( \Theta\left( \phi^{(K)}\left( \mathbf{h}_i^{(K-1)}, \bigoplus_{j \in N_i} \psi^{(K)}(\mathbf{h}_i^{(K-1)}, \mathbf{h}_j^{(K-1)}) \right) \right) \right) \right) \tag{3}$$

$$\mathbf{h}_i^{(l)} = \phi^{(l)}\left( \mathbf{h}_i^{(l-1)}, \bigoplus_{j \in N_i} \psi^{(l)}(\mathbf{h}_i^{(l-1)}, \mathbf{h}_j^{(l-1)}) \right) \quad l = 1, \dots, K \tag{4}$$

where $f$ is an interpretable classifier (e.g., single-layer network), $\boxplus$ is an interpretable piece-wise linear and permutation-invariant function (such as max or sum), $\Theta$ is a Gumbel-Softmax hard activation function, and $\mathbf{h}_i^0 = \mathbf{x}_i$. In this way, we can interpret the first part of the network as a feature extractor generating well-clustered latent representations from which concepts can be extracted. This approach is useful when we only care about the most complex neighborhoods/concepts. Another approach is to generate a hierarchical transparent architecture where each GNN layer is interpretable:

$$\hat{y}^{(l)} = f\left( \boxplus_{i \in K} \left( \Theta\left( \mathbf{h}_j^{(l)} \right) \right) \right) \qquad l = 1, \dots, K \tag{5}$$

In this case, we can interpret every single layer of our model with concepts of increasing complexity. The concepts extracted from the first layer represent subgraphs corresponding to the 1-hop neighborhood of a node, those extracted at the second layer will correspond to 2-hop neighborhoods, and so on. These hierarchical iGNNs can be useful to get insights into concepts with different granularities. By analyzing the concepts extracted at each layer, we gain a better understanding of the GNN inference and of the importance of different (sub)graph structures for the classification task.

### 3.2.3 Training

For the classification layer, the choice of the activation and loss functions for iGNNs depends on the nature of the task at hand and does not affect their interpretability. For classification tasks, we use standard activation functions such as softmax or sigmoid, along with standard loss functions like cross-entropy. For hierarchical iGNNs (HiGNNs), we apply the same loss function at each layer of the concept hierarchy, as their layered architecture enables intermediate supervisions. This ensures that each layer is doing its best to extract the most relevant concepts to solve the task. Internal losses can also be weighted differently to prioritize the formation of optimal concepts of a specific size, allowing the HiGNN to learn in a progressive and efficient way.

## 4 Experimental Analysis

### 4.1 Research questions

In this section we analyze the following research questions:

- **Generalization** - Can GNNs generalize when trained to predict universal algebra's properties? Can interpretable GNNs generalize as well?

- **Interpretability** - Do interpretable GNNs concepts empirically validate universal algebra's conjectures? How can concept-based explanations suggest novel conjectures?

### 4.2 Setup

**Baselines.** For our comparative study, we evaluate the performance of iGNNs and their hierarchical version against equivalent GNN models (i.e., having the same hyperparameters such as number layers, training epochs, and learning rate). For vanilla GNNs we resort to common practice replacing the Gumbel-Softmax with a standard leaky ReLU activation. We exclude from our main baselines prototype or concept-based GNNs pre-defining graph structures for explanations, as for most datasets these structures are unknown. Appendix C covers implementation details. We show more extensive results including local and post-hoc explanations in Appendix G.

**Evaluation.** We employ three quantitative metrics to assess a model's generalization and interpretability. We use the Area Under the Receiver Operating Characteristic (AUC ROC) curve to assess task generalization. We evaluate generalization under two different conditions: with independently and identically distributed train/test splits, and out-of-distribution by training on graphs up to eight nodes, while testing on graphs with more than eight nodes ("strong generalization" (41)). We further assess generalization under binary and multilabel settings (classifying 5 properties of a lattice at the same time). To evaluate interpretability, we use standard metrics such as completeness (44) and fidelity (37). Completeness[4] assesses the quality of the concept space on a global scale using an interpretable model to map concepts to tasks, while fidelity measures the difference in predictions obtained with an interpretable surrogate model and the original model. Finally, we evaluate the meaningfulness of our concept-based explanations by visualizing and comparing the generated concepts with ground truth lattices like e.g. $M_3$ and $N_5$, whose omission is known to be significant for modular and distributive properties. All metrics in our evaluation, across all experiments, are computed on test sets using 5 random seeds, and reported using the mean and 95% confidence interval.

## 5 Key Findings

### 5.1 Generalization

**iGNNs improve interpretability without sacrificing task accuracy (Figure 4).** Our experimental evaluation reveals that interpretable GNNs are able to strike a balance between completeness and fidelity, two crucial metrics that are used to assess generalization-interpretability trade-offs (37). We observe that the multilabel classification scenario, which requires models to learn a more varied and diverse set of concepts, is the most challenging and results in the lowest completeness scores on average. We also notice that the more challenging out-of-distribution scenario results in the lowest completeness and fidelity scores across all datasets. More importantly, our findings indicate that iGNNs achieve optimal fidelity scores, as their classification layer consists of a simple linear function of the learnt concepts which is intrinsically interpretable (38). On the contrary, interpretable surrogate models of black-box GNNs exhibit, as expected, lower fidelity scores, confirming analogous observations in the explainable AI literature (37; 38). In practice, this discrepancy between the original black-box predictions and the predictions obtained with an interpretable surrogate model questions the actual usefulness of black-boxes when interpretable alternatives achieve similar results in solving the problem at hand, as extensively discussed by Rudin (38). Overall, these results demonstrate how concept spaces are highly informative to solve universal algebra's tasks and how the interpretable graph layer may improve GNNs' interpretability without sacrificing task accuracy. We refer the reader to Appendix E for detailed discussion on quantitative analysis of concept space obtained by iGNNs under different generalization settings with comparisons to their black-box counterparts.

**GNNs strongly generalize on universal algebra's tasks (Figure 5).** Our experimental findings demonstrate the strong generalization capabilities of GNNs across the universal algebra tasks we designed. Indeed, we stress GNNs test generalization abilities by training the models on graphs of size up to $n$ (with $n$ ranging from 5 to 8), and evaluating their performance on much larger graphs of size up to 50. We designed this challenging experiment in order to understand the limits and robustness of interpretable GNNs when facing a significant data distribution shift from training to test data. Remarkably, iGNNs exhibit robust generalization abilities (similar to their black-box counterparts) when trained on graphs up to size 8 and tested on larger graphs. This evidence confirms the hypothesis that interpretable models can deliver reliable and interpretable predictions, as suggested by Rudin

---

[4]We assess the recall of the completeness as the datasets are very unbalanced towards the negative label.

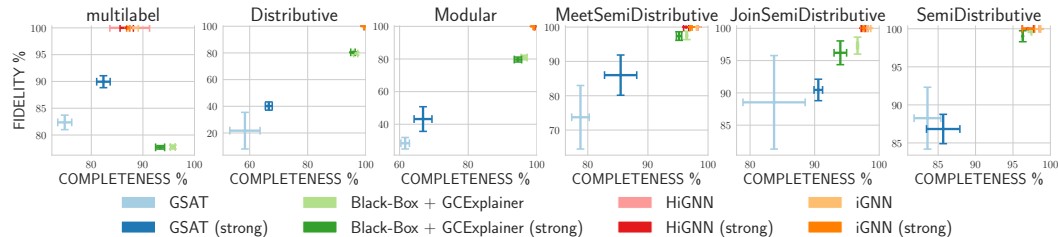

Figure 4: Accuracy-interpretability trade-off in terms of concept completeness (accuracy) and model fidelity (interpretability). iGNNs attain optimal fidelity as model inference is inherently interpretable, outmatching equivalent black-box GNNs. All models attain similar results in terms of completeness.

(38). However, we observe that black-box GNNs slightly outperform iGNNs when trained on even smaller lattices. We hypothesize that this is due to the more constrained architecture of iGNNs, which imposes tighter bounds on their expressiveness when compared to standard black-box GNNs.

Notably, training with graphs of size up to $5$ or $6$ significantly diminishes GNNs generalization in the tasks we designed. We hypothesize that this is due to the scarcity of non-distributive and non-modular lattices during training, but it may also suggest that some patterns of size $7$ and $8$ might be quite relevant to generalize

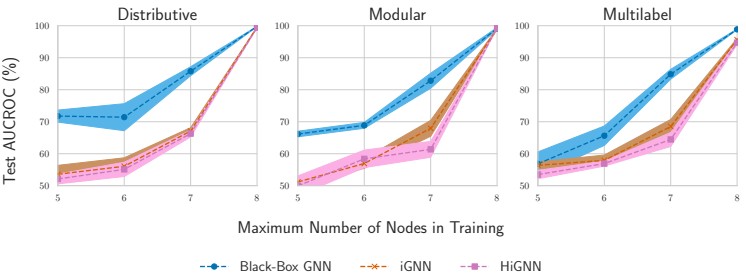

Figure 5: Strong generalization performance with respect to the maximum number of nodes used in training.

to larger graphs. Unfortunately, running generalization experiments with $n \leq 4$ was not possible since all such lattices trivially omitted $\mathbf{N}_5$ and $\mathbf{M}_3$. It is worth mentioning that GNNs performed well even in the challenging multilabel case, where they had to learn a wider and more diverse set of concepts and tasks. In all experiments, we observe a plateau of the AUC ROC scores for $n = 8$, thus suggesting that a training set including graphs of this size might be sufficient to learn the relevant patterns allowing the generalization to larger lattice varieties. For detailed numerical results across all tasks, we refer the reader to Table 1 in Appendix D. Overall, these results emphasize the potential of GNNs in addressing complex problems in universal algebra, providing an effective tool to handle lattices that are difficult to analyze manually with pen and paper.

## 5.2 Interpretability

**Concept-based explanations empirically validate universal algebra's conjectures (Figure 6).** We present empirical evidence to support the validity of theorems 2.3 and 2.4 by examining the concepts generated for modular and distributive tasks. For this investigation we leverage the interpretable structure of iGNNs.

Similarly to Ribeiro et al. (37), we visualize in Figure 6 the weights of our trained linear classifier representing the relevance of each concept. We remark that the visualization is limited to the (top-5) most negative weights, as we are interested in those concepts that negatively affect the prediction of a property. In the same plot, we also show the prototype of each concept represented by the 2-hop neighbor-

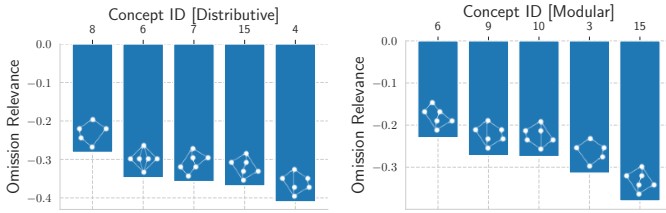

Figure 6: Ranking of relevant clusters of lattices (x-axis) according to the interpretable GNN linear classifier weights (y-axis, the lower the more relevant the cluster). $\mathbf{N}_5$ is always the most important sub-lattice to omit for modularity, while both $\mathbf{M}_3$ and $\mathbf{N}_5$ are relevant for distributivity, thus validating theorems 2.3 and 2.4.

hood of a node activating the concept, following a similar procedure as Ghorbani et al. (10); Magister et al. (28); Azzolin et al. (2). Using this visualization, we investigate the presence of certain concepts when classifying modular and distributive lattices. For the modularity task, our results show that the lattice $N_5$ appears among non-modular concepts, but is never found in modular lattices, while the lattice $M_3$ appears among both modular and non-modular concepts, which is consistent with Theorem 2.3. In the case of distributivity, we observe that both $M_3$ and $N_5$ are present among non-distributive concepts, and are never found in distributive lattices, which is also in line with Theorem 2.4. These findings provide a large-scale empirical evidence for the validity of theorems 2.3 and 2.4, and further demonstrate the effectiveness of graph neural networks in learning and analyzing lattice properties. Overall, these results highlight how interpretable GNNs can not only learn the properties of universal algebra but also identify structures that are unique to one type of lattice (e.g., non-modular) and absent from another (e.g., modular), thus providing human-interpretable explanations for what the models learn.

**Contrastive explanations highlight topological differences between properties of lattice varieties (Figure 7).** We leverage interpretable GNNs to analyze the key topological differences of classical lattice properties such as join and meet semi-distribuitivity characterized by relevant quasi-equations (cf. Appendix A.5). To this end, we visualize specific concept prototypes corresponding to lattices that are not meet semi-distributive against lattices that are meet semi-distributive.

We observe $N_5$ but not $M_3$ among the concepts of meet semi-distributive lattices, while we observe both $N_5$ and $M_3$ only in concepts that are not meet semi-distributive. This observation suggests that $N_5$ is not a key lattice for meet semi-distributive lattices, *unlike distributive lattices*. Furthermore, we find that the lattice pattern 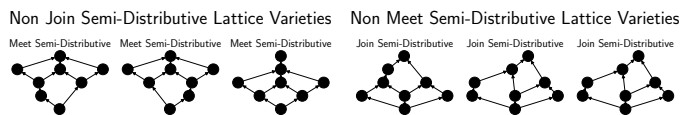 is relevant for non meet semidistributivity, while its dual is relevant for non join semidistributivity, thus empirically confirming the hypotheses of Jónsson and Rival (19). These findings are significant because they demonstrate how analyzing concepts in interpretable GNNs can provide universal algebraists with a powerful and automatic tool to formulate new conjectures based on identifying simple lattices that play a role in specific properties. By leveraging the power of interpretable GNNs, we may uncover previously unknown connections between different properties and identify new patterns and structures that could lead to the development of new conjectures and theorems in universal algebra, providing exciting opportunities for future research in universal algebra.

Figure 7: Contrastive explanations showing lattice varieties with a a pair of discording labels to highlight the key difference between join and meet semi-distributivity.

# 6 Discussion

**Relations with Graph Neural Network explainability.** Graph Neural Networks (GNNs,(39)) process relational data generating node representations by combining the information of a node with that of its neighbors, thanks to a general learning paradigm known as message passing (11). A number of post-hoc explainability techniques have been proposed to explain the reasoning of GNNs. Inspired by vision approaches (40; 37; 10), early explainability techniques focused on feature importance (35), while subsequent works aimed to extract local explanations (45; 27; 42) or global explanations using conceptual subgraphs by clustering the activation space (28; 46; 29; 22). However, all these techniques either rely on pre-defined subgraphs for explanations (which are often unknown in UA) or provide post-hoc explanations which may be brittle and unfaithful as extensively demonstrated by Rudin (38). On the contrary, our experiments show that iGNNs generate interpretable predictions according to Rudin (38) notion of interpretability via linear classifiers applied on sparse human-understandable concept representations.

**Limitations.** The approach proposed in this paper focuses on universal algebra conjectures characterized both algebraically and topologically. Our methodology is limited to finite lattices, which may not capture all relevant information about infinite algebraic structures. However, the insights gained from finite-lattice explanations can still provide valuable information regarding a given problem (albeit with potentially limited generalization). Moreover, our approach is restricted to topological

properties on graphs, while non-structural properties may require the adoption of other kinds of (interpretable) models.

**Broader impact and perspectives.** AI techniques are becoming increasingly popular for solving previously intractable mathematical problems and proposing new conjectures (23; 26; 7; 12). However, the use of modern AI methods in universal algebra was a novel and unexplored field until the development of the approach presented in this paper. To this end, our method uses interpretable graph networks to suggest graph structures that characterize relevant algebraic properties of lattices. With our approach, we empirically validated Dedekind (9) and Birkhoff (5) theorems on distributive and modular lattices, by recovering relevant lattices. This approach can be readily extended—beyond equational properties determined by the omission of a sublattice in a variety (43)—to any structural property of lattices, including the characterization of congruence lattices of algebraic varieties (1; 21; 31; 43). Our methodology can also be applied (beyond universal algebra) to investigate (almost) any mathematical property that can be topologically characterized on a graph, such as the classes of graphs/diagraphs with a fixed set of polymorphisms (25; 3; 32).

**Conclusion.** This paper presents the first-ever AI-assisted approach to investigate equational and topological conjectures in the field of universal algebra. To this end, we present a novel algorithm to generate datasets suitable for AI models to study equational properties of lattice varieties. While topological representations would enable the use of graph neural networks, the limited transparency and brittle explainability of these models hinder their use in validating existing conjectures or proposing new ones. For this reason, we introduce a novel neural layer to build fully interpretable graph networks to analyze the generated datasets. The results of our experiments demonstrate that interpretable graph networks: enhance interpretability without sacrificing task accuracy, strongly generalize when predicting universal algebra's properties, generate simple explanations that empirically validate existing conjectures, and identify subgraphs suggesting the formulation of novel conjectures. These promising results demonstrate the potential of our methodology, opening the doors of universal algebra to AI with far-reaching impact across all mathematical disciplines.

## Acknowledgments and Disclosure of Funding

This paper was supported by TAILOR and by HumanE-AI-Net projects funded by EU Horizon 2020 research and innovation programme under GA No 952215 and No 952026, respectively. This paper has been also supported by the Austrian Science Fund FWF project P33878 "Equations in Universal Algebra" and the European Union's Horizon 2020 research and innovation programme under grant agreement No 848077. This project has also received funding from the European Union's Horizon-MSCA-2021 research and innovation program under grant agreement No 101073307.

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

# A  Algebra definitions

## A.1  Formal defintions for Universal Algebra

Universal algebra is the field of mathematics that studies algebraic structures, which are defined as a set $A$ along with its own collection of operations. In this section, we recall some basic definitions and theorems from Burris and Sankappanavar (6); Day (8); Jonnson (17), about elements of universal algebra and lattice theory.

**Definition A.1. N-ary function** For a non-empty set $A$ and $n$ non-negative integer we define $A^0 = \{\emptyset\}$ and, for $n > 0$, $A^n$ is the set of n-tuples of elements from $A$. An $n$-ary *operation* on $A$ is any function $f$ from $A^n$ to $A$; $n$ is the *arity* of $f$. An operation $f$ on $A$ is called an $n$-ary operation if its arity is $n$.

**Definition A.2. Algebraic Structure** An *algebra* **A** is a pair $(A, F)$ where $A$ is a non-empty set called *universe* and $F$ is a set of finitary operations on $A$.

Apart from the operations on $A$, an algebra is further defined by axioms, that in the particular case of universal algebras are in the form of identities.

**Definition A.3.** A *lattice* **L** is an algebraic structure composed by a non-empty set $L$ and two binary operations $\vee$ and $\wedge$ satisfying the following axioms and their duals obtained exchanging $\vee$ and $\wedge$:

$$x \vee y \approx y \vee x \qquad \text{(commutativity)}$$
$$x \vee (y \vee z) \approx (x \vee y) \qquad \text{(associativity)}$$
$$x \vee x \approx x \qquad \text{(idempotency)}$$
$$x \approx x \vee (x \wedge y) \qquad \text{(absorption)}$$

**Theorem A.4** ((6)). *A partially ordered set $L$ is a lattice if and only if for every $a, b \in L$ both* supremum *and* infimum *of $\{a, b\}$ exist (in L) with $a \vee b$ being the supremum and $a \wedge b$ the infimum.*

**Definition A.5.** Let **L** be a lattice. Then **L** is *modular* (*distributive, $\vee$-semi-distributive, $\wedge$-semi-distributive*) if it satisfies the following:

$$x \leq y \rightarrow x \vee (y \wedge z) \approx y \wedge (x \vee z) \qquad \text{(modularity)}$$
$$x \vee (y \wedge z) \approx (x \vee y) \wedge (x \vee z) \qquad \text{(distributivity)}$$
$$x \vee y \approx x \vee z \rightarrow x \vee (y \wedge z) \approx x \vee y \qquad \text{($\vee$-semi-distributivity)}$$
$$x \wedge y \approx x \wedge z \rightarrow x \wedge (y \vee z) \approx x \wedge y \qquad \text{($\wedge$-semi-distributivity)}.$$

Furthermore a lattice **L** is semi-distributive if is both $\vee$-semi-distributive and $\wedge$-semi-distributive

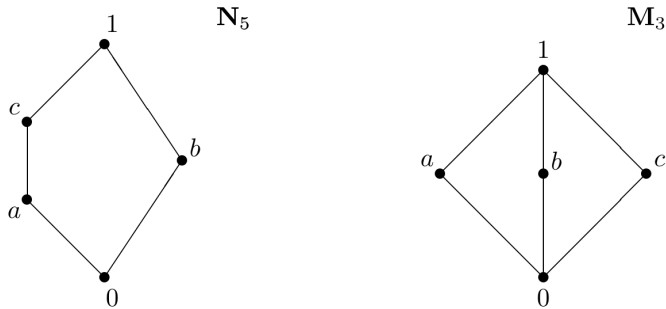

Figure 8: $\mathbf{N}_5$, a non-modular non-distributive and $\mathbf{M}_3$, a modular non-distributive lattice.

**Theorem A.6.** *If a lattice **L** is distributive, then **L** is also modular.*

*Proof.* By assuming $x \leq y$, we have $x \vee y = y$. Hence, from the distributive property we get:
$$x \vee (y \wedge z) \approx (x \vee y) \wedge (x \vee z) \approx y \wedge (x \vee z)$$

$\square$

**Definition A.7. Congruence Lattice**

An *equivalence relation* on a set $A$ is a binary relation $\sim$ that satisfies three properties: reflexivity, symmetry, and transitivity.

Reflexivity: For every element $a$ in $A$, $a$ is related to itself, denoted as $a \sim a$;

Symmetry: For any elements $a$ and $b$ in $A$, if $a \sim b$, then $b \sim a$;

Transitivity: For any elements $a$, $b$, and $c$ in $A$, if $a \sim b$ and $b \sim c$, then $a \sim c$.

In other words, an equivalence relation partitions the set $A$ into subsets, called *equivalence classes*, such that elements within the same class are equivalent to each other under the relation $\sim$.

Let $\mathbf{A}$ be an algebra. A *congruence* $\theta$ of $\mathbf{A}$ is a equivalent relation on $A$, that is compatible with the operations of $\mathbf{A}$. Formally, for every $n$-ary operation $f$ of $\mathbf{A}$: if $(a_1, b_1), (a_2, b_2), \dots, (a_n, b_n) \in \theta$, then $(f(a_1, a_2, \dots, a_n), f(b_1, b_2, \dots, b_n)) \in \theta$. For every algebra $\mathbf{A}$ on the set $A$, the identity relation on $A$, and $A \times A$ are trivial congruences. An algebra with no other congruences is called *simple*. Let $\mathrm{Con}(\mathbf{A})$ be the set of congruences on the algebra $\mathbf{A}$. Since congruences are closed under intersection, we can define a meet operation: $\wedge : \mathrm{Con}(\mathbf{A}) \times \mathrm{Con}(\mathbf{A}) \to \mathrm{Con}(\mathbf{A})$ by simply taking the intersection of the congruences $E_1 \wedge E_2 = E_1 \cap E_2$. Congruences are not closed under union, however we can define the following closure operator of a binary relation $E$, with respect to a fixed algebra $\mathbf{A}$, such that its image is congruence: $\langle E \rangle_{\mathbf{A}} = \bigcap \{F \in \mathrm{Con}(\mathbf{A}) \mid E \subseteq F\}$. Note that the closure of a binary relation is a congruence and thus depends on the operations in $\mathbf{A}$, not just on the base set. Now define $\vee : \mathrm{Con}(\mathbf{A}) \times \mathrm{Con}(\mathbf{A}) \to \mathrm{Con}(\mathbf{A})$ as $E_1 \vee E_2 = \langle E_1 \cup E_2 \rangle_{\mathbf{A}}$. For every algebra $\mathbf{A}$, $(\mathrm{Con}(\mathbf{A}), \wedge, \vee)$ with the two operations defined above forms a lattice, called the *congruence lattice* of $\mathbf{A}$.

A *type* $\mathcal{F}$ is defined as a set of operation symbols along with their respective arities. Each operation symbol represents a specific operation that can be performed on the elements of the algebraic system. To refer to the specific operation performed by a given symbol $f$ on an algebra $\mathbf{A}$ of type $\mathcal{F}$, we denote it as $f^{\mathbf{A}}$. This notation allows us to differentiate and access the particular operation carried out by $f$ within the context of $\mathbf{A}$.

**Definition A.8. Subalgebra** Let $\mathbf{A}$ and $\mathbf{B}$ be two algebras of the same type. Then $\mathbf{B}$ is a *subalgebra* of $\mathbf{A}$ if $B \subseteq A$ and every fundamental operation of $\mathbf{B}$ is the restriction of the corresponding operation of $\mathbf{A}$, i.e., for each function symbol $f$, $f^{\mathbf{B}}$ is $f^{\mathbf{A}}$ restricted to $\mathbf{B}$.

**Definition A.9. Homomorphic image** Suppose $\mathbf{A}$ and $\mathbf{B}$ are two algebras of the same type $\mathcal{F}$, i.e. for each operation of $\mathbf{A}$, there exists a corresponding operation $\mathbf{B}$ with the same arity, and vice versa. A mapping $\alpha : A \to B$ is called a *homomorphism* from $\mathbf{A}$ to $\mathbf{B}$ if

$$\alpha f^{\mathbf{A}}(a_1, \dots, a_n) = f^{\mathbf{B}}(\alpha a_1, \dots, \alpha a_n)$$

for each n-ary $f$ in $\mathcal{F}$ and each sequence $a_1, \dots, a_n$ from $\mathbf{A}$. If, in addition, the mapping $\alpha$ is onto then $\mathbf{B}$ is said to be a *homomorphic image* of $\mathbf{A}$.

**Definition A.10. Direct product** Let $\mathbf{A}_1$ and $\mathbf{A}_2$ be two algebras of the same type $\mathcal{F}$. We define the direct product $\mathbf{A}_1 \times \mathbf{A}_2$ to be the algebra whose universe is the set $A_1 \times A_2$, and such that for $f \in \mathcal{F}$ and $a_i \in A_1$, $a_i' \in A_2$, $1 \le i \le n$,

$$f^{\mathbf{A}_1 \times \mathbf{A}_2}(\langle a_1, a_1' \rangle, \dots, \langle a_n, a_n' \rangle) = \langle f^{\mathbf{A}_1}(a_1, \dots, a_n), f^{\mathbf{A}_2}(a_1', \dots, a_n') \rangle$$

The collection of algebraic structures defined by equational laws are called varieties. (15)

**Definition A.11. Variety** A nonempty class K of algebras of type $\mathcal{F}$ is called a *variety* if it is closed under subalgebras, homomorphic images, and direct products.

# B    Algorithm 1 details

In the following, we report some technical details on how the dataset generator sketched in Algorithm 1 is actually implemented. First, notice that a brute-force approach is infeasible for large lattices, as given a set of $n$ nodes, the number of binary relations on this set is $2^{n^2}$. To cope with this issue, first

for each candidate lattice $\mathbf{L}$ we consider a squared $n \times n$ matrix $\leq_L$ representing its order that has 1 value in a position $(i, j)$ if and only if the element $i$ is less or equal to the element $j$ in $\mathbf{L}$ (i.e. $i \leq_L j$) and 0 otherwise. Then we constraint each matrix to have 1 in the diagonal (reflexivity), 0 in each $(i, j)$ with $j < i$, where "$<$" denotes denotes the order on $\mathbb{N}$ (this choice both prunes the majority of isomorphic lattices and yields anti-symmetricity). All the other pairs of elements $(i, j)$ can either be such that $i \leq_L j$ or incomparable (i.e. $i \not\leq_L j$ and $j \not\leq_L i$), and we consider all these possible cases. Finally, we apply matrix multiplications to get a transitive closure of the order relation (convergence guaranteed in at most $n - 2$ steps) and hence $\leq_L$ represents a partial order. To assure that the order represents a lattice, we have to check that each pair of elements $(i, j)$ admits a (unique!) infimum and supremum. This step and checking lattice equational properties are implemented tensorially to leverage GPU quicker computations, hence being particularly advantageous when the dimensions of the lattices is such that the computational cost of Algorithm 1 surpasses the overhead of GPU communication. Finally, we notice that even avoiding the isomorphic lattices, for $n = 18$ there are around 165Bn non-isomorphic different lattices (13). This is why we sampled a fixed number of lattices as $n$ increases, e.g. 20 samples for cardinality after a certain threshold, instead of keeping generating all the possible lattices for each value of $n$, which is not particularly relevant for our task. Whereas it allows us to study the strong generalization capability of GNNs trained on e.g. up to $n = 8$ nodes and then evaluated on lattices of higher dimensions, e.g. $n = 50$ nodes (see Figure 5). Notice that checking e.g. the distributivity for $n = 50$ is deterministic but requires checking $50^3$ identities.

**Running time.** The running time of the algorithm increases polynomially in the size $n$ of the given lattice (it is $O(n^3)$ for checking each of the equational properties, e.g. distributivity/modularity, and $O(n^4)$ to check if a candidate binary relation is actually a lattice). In a machine with a single quad-core CPU, it requires 20 minutes to generate all the lattices up to $N = 8$ and sampling $n_s = 20$ lattices for $n \in [9, 50]$ (the dataset we used in the paper).

## C   Baselines' details

In practice, we train all models using eight message passing layers and different embedding sizes ranging from 16 to 64. We train all models for 200 epochs with a learning rate of 0.001. For interpretable models, we set the Gumbel-Softmax temperature to the default value of 1 and the activation behavior to "hard," which generates one-hot encoded embeddings in the forward pass, but computes the gradients using the soft scores. For the hierarchical model, we set the internal loss weight to 0.1 (to score it roughly 10% less w.r.t. the main loss). Overall, our selection of baselines aims at embracing a wide set of training setups and architectures to assess the effectiveness and versatility of GNNs for analyzing lattice properties in universal algebra. To demonstrate the robustness of our approach, we implemented different types of message passing layers, including graph convolution and GIN.

## D   Generalization results details

Here we report the raw numbers for the weak and strong generalization results reported as a figure in the main paper. The results are obtained by setting maximum lattice size to 8 in training and using lattices of size 9 or larger during evaluation. All models provide near perfect performance for binary classification and perform slightly worse but still very competitively for multi-label classification. This demonstrates the strong generalization capability of GNNs for universal algebra tasks, and may be an ideal starting point to finding new relevant patterns in UA properties.

In Table 1 we also include a quantitative comparison with GSAT (30), and observe that GSAT and iGNNs obtain comparable results in terms of task generalization and concept completeness (cf. Table 4) when trained on the proposed UA's tasks.

## E   Concept completeness and purity

Our experimental results (Tables 2 & 4) demonstrate that interpretable GNNs produce concepts with high completeness and low purity, which are standard quantitative metrics used to evaluate the quality of concept-based approaches. Completeness score is the accuracy of a classifier, such as decision

Table 1: Generalization performance of different graph neural models in solving universal algebra's tasks. Values represents the mean and the standard error of the mean of the area under the receiver operating curve (AUCROC, %).

| | weak generalization | | | | strong generalization | | | |
|---|---|---|---|---|---|---|---|---|
| | GCExplainer | GSAT | iGNN | HiGNN | GCExplainer | GSAT | iGNN | HiGNN |
| Distributive | 99.80 ± 0.04 | 96.73 ± 0.44 | 99.56 ± 0.12 | 99.45 ± 0.06 | 99.51 ± 0.20 | 97.26 ± 0.30 | 99.44 ± 0.05 | 99.42 ± 0.04 |
| Join Semi Distributive | 99.49 ± 0.02 | 98.33 ± 0.06 | 98.31 ± 0.15 | 98.28 ± 0.04 | 98.77 ± 0.15 | 97.76 ± 0.07 | 97.50 ± 0.14 | 97.48 ± 0.14 |
| Meet Semi Distributive | 99.52 ± 0.04 | 98.36 ± 0.04 | 98.19 ± 0.06 | 98.25 ± 0.08 | 98.90 ± 0.03 | 97.85 ± 0.10 | 97.18 ± 0.14 | 96.89 ± 0.37 |
| Modular | 99.77 ± 0.02 | 96.62 ± 0.31 | 99.18 ± 0.11 | 99.35 ± 0.09 | 99.32 ± 0.22 | 96.35 ± 0.31 | 99.21 ± 0.14 | 99.11 ± 0.22 |
| Semi Distributive | 99.66 ± 0.03 | 98.76 ± 0.04 | 98.57 ± 0.02 | 98.50 ± 0.06 | 99.19 ± 0.04 | 98.14 ± 0.12 | 97.28 ± 0.48 | 96.88 ± 0.47 |
| Multi Label | 99.60 ± 0.02 | 95.00 ± 0.52 | 96.32 ± 0.34 | 95.98 ± 0.50 | 98.62 ± 0.43 | 94.45 ± 0.38 | 95.29 ± 0.55 | 95.27 ± 0.32 |

tree, which takes concepts as inputs and predicts a label. Purity score is the number of graph edits, such as node/edge addition/eliminations, necessary to match two graphs in a cluster. A concept space is said to be pure if the purity score is zero.

We employ decision tree as the classifier, but compute recall instead of accuracy to calculate completeness score since the datasets are heavily unbalanced towards the negative labels. We compute purity scores for each cluster and report the average of those scores as the final purity score. Our approach achieves at least 73% and up to 87% recall, which shows that our interpretable models consistently avoid false negatives in the abundance of negative labels. We obtain around 3-4 purity scores, which suggests that our interpretable models extract relatively pure concept spaces in the presence of large lattices.

Furthermore, the hierarchical structure of interpretable GNNs enables us to evaluate the quality of intermediate concepts layer by layer. This hierarchy provides insights into why we may need more layers, and it can be used as a valuable tool to find the optimal setup and tune the size of the architecture. Additionally, it can also be used to compare the quality of concepts at different layers of the network. To that end, we compare the purity scores of the concept spaces obtained by the second layer and the final layer of HiGNN. As shown in Table 3, deeper layers may produce higher quality concepts for distributivity and join semi-distributivity whereas earlier layers may result in more reliable concepts for the remaining properties. Overall, these results quantitatively assess and validate the high quality of the concepts learned by the interpretable GNNs, highlighting the effectiveness of this approach for learning and analyzing complex algebraic structures.

Table 2: Concept purity scores of graph neural models in solving universal algebra's tasks. Lower is better.

| | WEAK PURITY | | | STRONG PURITY | | |
|---|---|---|---|---|---|---|
| | GCExplainer | iGNN | HiGNN | GCExplainer | iGNN | HiGNN |
| Distributive | 3.30 ± 0.36 | 3.64 ± 0.30 | 3.09 ± 0.56 | 3.29 ± 0.38 | 4.00 ± 0.77 | 4.15 ± 0.67 |
| Join Semi Distributive | 2.38 ± 0.37 | 3.96 ± 0.51 | 3.74 ± 0.62 | 3.45 ± 0.34 | 3.98 ± 0.68 | 4.29 ± 0.61 |
| Meet Semi Distributive | 3.24 ± 0.63 | 3.55 ± 0.62 | 3.39 ± 0.29 | 3.36 ± 0.32 | 4.25 ± 0.39 | 4.97 ± 0.44 |
| Modular | 3.10 ± 0.35 | 3.50 ± 0.46 | 4.44 ± 0.56 | 3.14 ± 0.24 | 3.19 ± 1.01 | 4.25 ± 0.69 |
| Semi Distributive | 2.84 ± 0.51 | 3.70 ± 0.54 | 4.11 ± 0.46 | 3.70 ± 0.55 | 3.92 ± 0.28 | 4.08 ± 0.85 |

Table 3: Concept purity scores of different layers of HiGNN. Lower is better.

| | WEAK PURITY | | STRONG PURITY | |
|---|---|---|---|---|
| | **2nd Layer** | **Last Layer** | **2nd Layer** | **Last Layer** |
| **Distributive** | 3.26 ± 0.43 | 3.09 ± 0.56 | 4.66 ± 0.98 | 4.15 ± 0.67 |
| **Join Semi Distributive** | 4.25 ± 0.69 | 3.74 ± 0.62 | 4.30 ± 0.39 | 4.29 ± 0.61 |
| **Meet Semi Distributive** | 3.64 ± 0.39 | 3.39 ± 0.29 | 4.41 ± 0.27 | 4.97 ± 0.44 |
| **Modular** | 3.89 ± 0.63 | 4.44 ± 0.56 | 4.19 ± 0.56 | 4.25 ± 0.69 |
| **Semi Distributive** | 3.55 ± 0.58 | 4.11 ± 0.46 | 3.16 ± 0.59 | 4.08 ± 0.85 |

Table 4: Concept completeness scores of graph neural models in solving universal algebra's tasks. Higher is better.

| | WEAK COMPLETENESS | | | | STRONG COMPLETENESS | | | |
|---|---|---|---|---|---|---|---|---|
| | GCExplainer | iGNN | GSAT | HiGNN | GCExplainer | iGNN | GSAT | HiGNN |
| Distributive | 96.53 ± 0.72 | 99.54 ± 0.13 | 58.36 ± 5.22 | 99.42 ± 0.09 | 95.61 ± 0.76 | 99.48 ± 0.06 | 66.59 ± 1.19 | 99.46 ± 0.06 |
| Join Semi-Distributive | 96.64 ± 0.14 | 98.45 ± 0.25 | 83.72 ± 4.82 | 98.19 ± 0.11 | 93.98 ± 0.97 | 97.59 ± 0.13 | 90.57 ± 0.66 | 97.51 ± 0.31 |
| Meet Semi-Distributive | 96.46 ± 0.11 | 98.17 ± 0.11 | 78.78 ± 1.45 | 98.30 ± 0.03 | 95.18 ± 0.44 | 97.20 ± 0.14 | 85.46 ± 2.68 | 96.36 ± 0.43 |
| Modular | 96.28 ± 0.86 | 99.08 ± 0.03 | 61.62 ± 1.24 | 99.40 ± 0.12 | 94.49 ± 1.08 | 99.10 ± 0.20 | 66.83 ± 2.60 | 99.33 ± 0.07 |
| SemiDistributive | 97.47 ± 0.04 | 98.62 ± 0.08 | 83.58 ± 1.79 | 98.57 ± 0.09 | 96.31 ± 0.02 | 97.07 ± 0.81 | 85.68 ± 2.23 | 97.00 ± 0.75 |
| Multilabel | 95.79 ± 0.47 | 88.21 ± 0.88 | 74.85 ± 1.35 | 87.44 ± 3.85 | 93.33 ± 0.87 | 87.16 ± 0.31 | 82.35 ± 1.30 | 86.86 ± 1.29 |

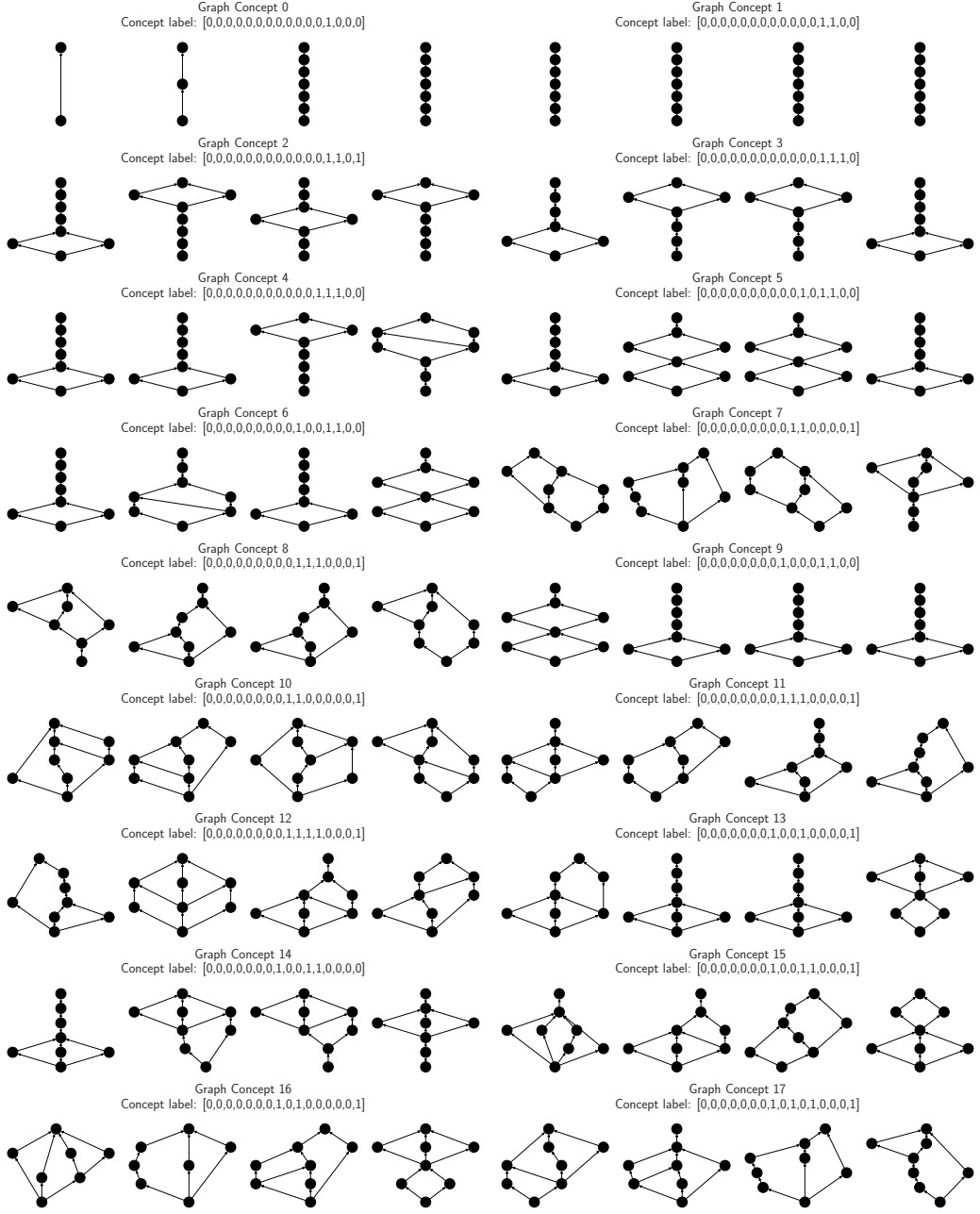

Figure 9: Examples of graph concepts.

# F Concept visualization

Figure 9 visualizes 18 randomly sampled graph concepts (out of the 7896 graph concepts represented by different graph encodings) following the visualization procedure introduced by (28). The figure shows for each concept an example of four (randomly sampled) graphs having the same concept label in the 7-th layer of the hierarchical iGNN trained on the multilabel dataset. Graphs belonging to the same concept show a coherency in their structure and similar patterns. These patterns represent the knowledge extracted and discovered by the hierarchical iGNN.

# G    Explanations of post-hoc explainers

We compared our Explainable Hierarchical GNN against a standard explainer (namely GNNExplainer (45)) to further support our results. GNNExplainer is the first general, model-agnostic approach for providing interpretable explanations for predictions of any GNN-based model on any graph-based machine learning task and it is widely used in the scientific community as one of the staple explainers in GNN's XAI. In this particular setting, GNNExplainer was configured as follows: model-wise explanation on multiclass-node level classification task, with HiGNN as the model of choice, and GNNExplainer as the desired algorithm, trained for 200 epochs. The explainer takes as input a single graph in the dataset and outputs and explanation for its classification. GNNExplainer will enforce a classification based on the presence or omission of $M_3$ and/or $N_5$ and it is possible to visualize the subgraph that lead to this classification by leveraging the `visualize_graph` function. By doing this, we retrieve the following visualizations:

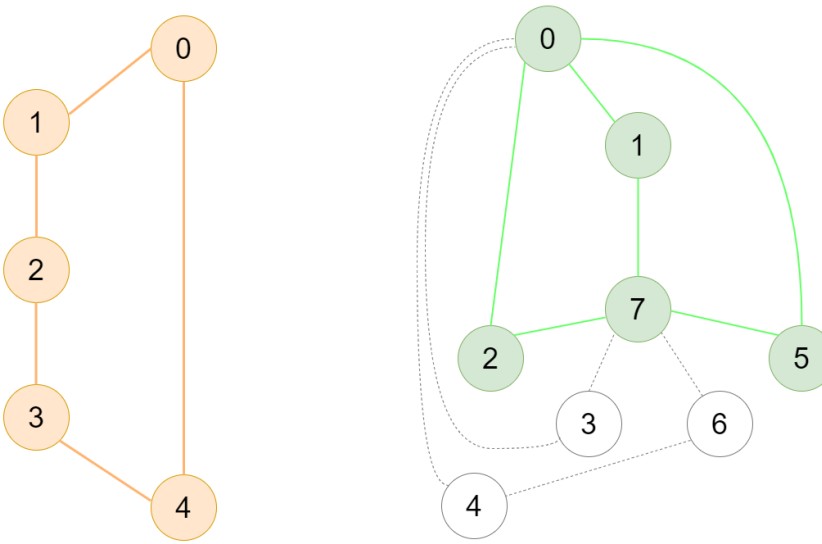

Figure 10: Visualizations obtained with GNNExplainer on weak distributive generalization (on the left) and strong multiclass generalization (on the right)

On the right, the substructure identified as $N5$ by `GNNExplainer` which lead to the classification of said graph as non modular and non distributive. On the right, in green $M3$. Our hierarchical model arrives to the same conclusions as the standard explainer but can also be augmented with a standard explainer.

# H    Code, Licences, Resources

**Libraries**    For our experiments, we implemented all baselines and methods in Python 3.7 and relied upon open-source libraries such as PyTorch 1.11 (33) (BSD license) and Scikit-learn (34) (BSD license). To produce the plots seen in this paper, we made use of Matplotlib 3.5 (BSD license). We will release all of the code required to recreate our experiments in an MIT-licensed public repository.

**Resources**    All of our experiments were run on a private machine with 8 Intel(R) Xeon(R) Gold 5218 CPUs (2.30GHz), 64GB of RAM, and 2 Quadro RTX 8000 Nvidia GPUs. We estimate that approximately 100-GPU hours were required to complete all of our experiments.

