# A    Algebra definitions

## A.1    Formal defintions for Universal Algebra

Universal algebra is the field of mathematics that studies algebraic structures, which are defined as a set $A$ along with its own collection of operations. In this section, we recall some basic definitions and theorems from Burris and Sankappanavar (6); Day (8); Jonnson (17), about elements of universal algebra and lattice theory.

**Definition A.1. N-ary function** For a non-empty set $A$ and $n$ non-negative integer we define $A^0 = \{\emptyset\}$ and, for $n > 0$, $A^n$ is the set of n-tuples of elements from $A$. An $n$-ary *operation* on $A$ is any function $f$ from $A^n$ to $A$; $n$ is the *arity* of $f$. An operation $f$ on $A$ is called an $n$-ary operation if its arity is $n$.

**Definition A.2. Algebraic Structure** An *algebra* **A** is a pair $(A, F)$ where $A$ is a non-empty set called *universe* and $F$ is a set of finitary operations on $A$.

Apart from the operations on $A$, an algebra is further defined by axioms, that in the particular case of universal algebras are in the form of identities.

**Definition A.3.** A *lattice* **L** is an algebraic structure composed by a non-empty set $L$ and two binary operations $\lor$ and $\land$ satisfying the following axioms and their duals obtained exchanging $\lor$ and $\land$:

$$x \lor y \approx y \lor x \qquad \text{(commutativity)}$$
$$x \lor (y \lor z) \approx (x \lor y) \qquad \text{(associativity)}$$
$$x \lor x \approx x \qquad \text{(idempotency)}$$
$$x \approx x \lor (x \land y) \qquad \text{(absorption)}$$

**Theorem A.4** ((6))**.** *A partially ordered set $L$ is a lattice if and only if for every $a, b \in L$ both* supremum *and* infimum *of $\{a, b\}$ exist (in L) with $a \lor b$ being the supremum and $a \land b$ the infimum.*

**Definition A.5.** Let **L** be a lattice. Then **L** is *modular* (*distributive, $\lor$-semi-distributive, $\land$-semi-distributive*) if it satisfies the following:

$$x \leq y \to x \lor (y \land z) \approx y \land (x \lor z) \qquad \text{(modularity)}$$
$$x \lor (y \land z) \approx (x \lor y) \land (x \lor z) \qquad \text{(distributivity)}$$
$$x \lor y \approx x \lor z \to x \lor (y \land z) \approx x \lor y \qquad \text{($\lor$-semi-distributivity)}$$
$$x \land y \approx x \land z \to x \land (y \lor z) \approx x \land y \qquad \text{($\land$-semi-distributivity).}$$

Furthermore a lattice **L** is semi-distributive if is both $\lor$-semi-distributive and $\land$-semi-distributive

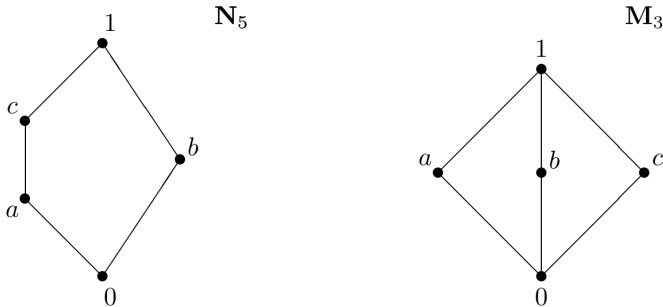

Figure 8: $\mathbf{N}_5$, a non-modular non-distributive and $\mathbf{M}_3$, a modular non-distributive lattice.

**Theorem A.6.** *If a lattice **L** is distributive, then **L** is also modular.*

*Proof.* By assuming $x \leq y$, we have $x \lor y = y$. Hence, from the distributive property we get:

$$x \lor (y \land z) \approx (x \lor y) \land (x \lor z) \approx y \land (x \lor

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

** | $99.80 \pm 0.04$ | $96.73 \pm 0.44$ | $99.56 \pm 0.12$ | $99.45 \pm 0.06$ | $99.51 \pm 0.20$ | $97.26 \pm 0.30$ | $99.44 \pm 0.05$ | $99.42 \pm 0.04$ |
| **Join Semi Distributive** | $99.49 \pm 0.02$ | $98.33 \pm 0.06$ | $98.31 \pm 0.15$ | $98.28 \pm 0.04$ | $98.77 \pm 0.15$ | $97.76 \pm 0.07$ | $97.50 \pm 0.14$ | $97.48 \pm 0.14$ |
| **Meet Semi Distributive** | $99.52 \pm 0.04$ | $98.36 \pm 0.04$ | $98.19 \pm 0.06$ | $98.25 \pm 0.08$ | $98.90 \pm 0.03$ | $97.85 \pm 0.10$ | $97.18 \pm 0.14$ | $96.89 \pm 0.37$ |
| **Modular** | $99.77 \pm 0.02$ | $96.62 \pm 0.31$ | $99.18 \pm 0.11$ | $99.35 \pm 0.09$ | $99.32 \pm 0.22$ | $96.35 \pm 0.31$ | $99.21 \pm 0.14$ | $99.11 \pm 0.22$ |
| **Semi Distributive** | $99.66 \pm 0.03$ | $98.76 \pm 0.04$ | $98.57 \pm 0.02$ | $98.50 \pm 0.06$ | $99.19 \pm 0.04$ | $98.14 \pm 0.12$ | $97.28 \pm 0.48$ | $96.88 \pm 0.47$ |
| **Multi Label** | $99.60 \pm 0.02$ | $95.00 \pm 0.52$ | $96.32 \pm 0.34$ | $95.98 \pm 0.50$ | $98.62 \pm 0.43$ | $94.45 \pm 0.38$ | $95.29 \pm 0.55$ | $95.27 \pm 0.32$ |

tree, which takes concepts as inputs and predicts a label. Purity score is the number of graph edits, such as node/edge addition/eliminations, necessary to match two graphs in a cluster. A concept space is said to be pure if the purity score is zero.

We employ decision tree as the classifier, but compute recall instead of accuracy to calculate completeness score since the datasets are heavily unbalanced towards the negative labels. We compute purity scores for each cluster and report the average of those scores as the final purity score. Our approach achieves at least 73% and up to 87% recall, which shows that our interpretable models consistently avoid false negatives in the abundance of negative labels. We obtain around 3-4 purity scores, which suggests that our interpretable models extract relatively pure concept spaces in the presence of large lattices.

Furthermore, the hierarchical structure of interpretable GNNs enables us to evaluate the quality of intermediate concepts layer by layer. This hierarchy provides insights into why we may need more layers, and it can be used as a valuable tool to find the optimal setup and tune the size of the architecture. Additionally, it can also be used to compare the quality of concepts at different layers of the network. To that end, we compare the purity scores of the concept spaces obtained by the second layer and the final layer of HiGNN. As shown in Table 3, deeper layers may produce higher quality concepts for distributivity and join semi-distributivity whereas earlier layers may result in more reliable concepts for the remaining properties. Overall, these results quantitatively assess and validate the high quality of the concepts learned by the interpretable GNNs, highlighting the effectiveness of this approach for learning and analyzing complex algebraic structures.

Table 2: Concept purity scores of graph neural models in solving universal algebra's tasks. Lower is better.

| | WEAK PURITY | | | STRONG PURITY | | |
| | GCExplainer | iGNN | HiGNN | GCExplainer | iGNN | HiGNN |
|---|---|---|---|---|---|---|
| **Distributive** | $3.30 \pm 0.36$ | $3.64 \pm 0.30$ | $3.09 \pm 0.56$ | $3.29 \pm 0.38$ | $4.00 \pm 0.77$ | $4.15 \pm 0.67$ |
| **Join Semi Distributive** | $2.38 \pm 0.37$ | $3.96 \pm 0.51$ | $3.74 \pm 0.62$ | $3.45 \pm 0.34$ | $3.98 \pm 0.68$ | $4.29 \pm 0.61$ |
| **Meet Semi Distributive** | $3.24 \pm 0.63$ | $3.55 \pm 0.62$ | $3.39 \pm 0.29$ | $3.36 \pm 0.32$ | $4.25 \pm 0.39$ | $4.97 \pm 0.44$ |
| **Modular** | $3.10 \pm 0.35$ | $3.50 \pm 0.46$ | $4.44 \pm 0.56$ | $3.14 \pm 0.24$ | $3.19 \pm 1.01$ | $4.25 \pm 0.69$ |
| **Semi Distributive** | $2.84 \pm 0.51$ | $3.70 \pm 0.54$ | $4.11 \pm 0.46$ | $3.70 \pm 0.55$ | $3.92 \pm 0.28$ | $4.08 \pm 0.85$ |

Table 3: Concept purity scores of different layers of HiGNN. Lower is better.

| | WEAK PURITY | | STRONG PURITY | |
| | **2nd Layer** | **Last Layer** | **2nd Layer** | **Last Layer** |
|---|---|---|---|---|
| **Distributive** | $3.26 \pm 0.43$ | $3.09 \pm 0.56$ | $4.66 \pm 0.98$ | $4.15 \pm 0.67$ |
| **Join Semi Distributive** | $4.25 \pm 0.69$ | $3.74 \pm 0.62$ | $4.30 \pm 0.39$ | $4.29 \pm 0.61$ |
| **Meet Semi Distributive** | $3.64 \pm 0.39$ | $3.39 \pm 0.29$ | $4.41 \pm 0.27$ | $4.97 \pm 0.44$ |
| **Modular** | $3.89 \pm 0.63$ | $4.44 \pm 0.56$ | $4.19 \pm 0.56$ | $4.25 \pm 0.69$ |
| **Semi Distributive** | $3.55 \pm 0.58$ | $4.11 \pm 0.46$ | $3.16 \pm 0.59$ | $4.08 \pm 0.85$ |

Table 4: Concept completeness scores of graph neural models in solving universal algebra's tasks. Higher is better.

| | WEAK COMPLETENESS | | | | STRONG COMPLETENESS | | | |
| | GCExplainer | iGNN | GSAT | HiGNN | GCExplainer | iGNN | GSAT | HiGNN |
|---|---|---|---|---|---|---|---|---|
| Distributive | $96.53 \pm 0.72$ | $99.54 \pm 0.13$ | $58.36 \pm 5.22$ | $99.42 \pm 0.09$ | $95.61 \pm 0.76$ | $99.48 \pm 0.06$ | $66.59 \pm 1.19$ | $99.46 \pm 0.06$ |
| Join Semi-Distributive | $96.64 \pm 0.14$ | $98.45 \pm 0.25$ | $83.72 \pm 4.82$ | $98.19 \pm 0.11$ | $93.98 \pm 0.97$ | $97.59 \pm 0.13$ | $90.57 \pm 0.66$ | $97.51 \pm 0.31$ |
| Meet Semi-Distributive | $96.46 \pm 0.11$ | $98.17 \pm 0.11$ | $78.78 \pm 1.45$ | $98.30 \pm 0.03$ | $95.18 \pm 0.44$ | $97.20 \pm 0.14$ | $85.46 \pm 2.68$ | $96.36 \pm 0.43$ |
| Modular | $96.28 \pm 0.86$ | $99.08 \pm 0.03$ | $61.62 \pm 1.24$ | $99.40 \pm 0.12$ | $94.49 \pm 1.08$ | $99.10 \pm 0.20$ | $66.83 \pm 2.60$ | $99.33 \pm 0.07$ |
| SemiDistributive | $97.47 \pm 0.04$ | $98.62 \pm 0.08$ | $83.58 \pm 1.79$ | $98.57 \pm 0.09$ | $96.31 \pm 0.02$ | $97.07 \pm 0.81$ | $85.68 \pm 2.23$ | $97.00 \pm 0.75$ |
| Multilabel | $95.79 \pm 0.47$ | $88.21 \pm 0.88$ | $74.85 \pm 1.35$ | $87.44 \pm 3.85$ | $93.33 \pm 0.87$ | $87.16 \pm 0.31$ | $82.35 \pm 1.30$ | $86.86 \pm 1.29$ |

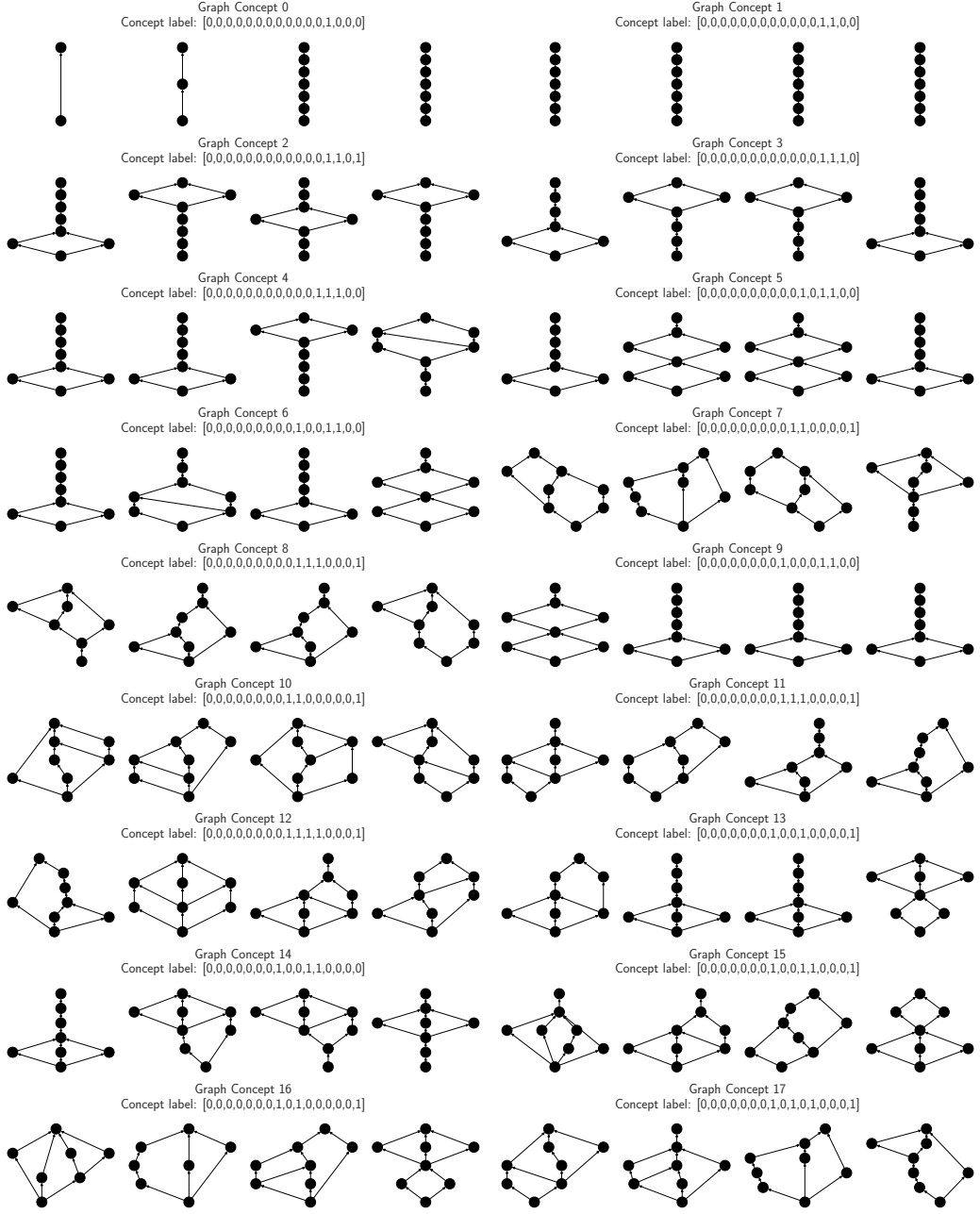

Figure 9: Examples of graph concepts.

# F    Concept visualization

Figure 9 visualizes 18 randomly sampled graph concepts (out of the 7896 graph concepts represented by different graph encodings) following the visualization procedure introduced by (28). The figure shows for each concept an example of four (randomly sampled) graphs having the same concept label in the 7-th layer of the hierarchical iGNN trained on the multilabel dataset. Graphs belonging to the same concept show a coherency in their structure and similar patterns. These patterns represent the knowledge extracted and discovered by the hierarchical iGNN.

# G   Explanations of post-hoc explainers

We compared our Explainable Hierarchical GNN against a standard explainer (namely GNNExplainer (44)) to further support our results. GNNExplainer is the first general, model-agnostic approach for providing interpretable explanations for predictions of any GNN-based model on any graph-based machine learning task and it is widely used in the scientific community as one of the staple explainers in GNN's XAI. In this particular setting, GNNExplainer was configured as follows: model-wise explanation on multiclass-node level classification task, with HiGNN as the model of choice, and GNNExplainer as the desired algorithm, trained for 200 epochs. The explainer takes as input a single graph in the dataset and outputs and explanation for its classification. GNNExplainer will enforce a classification based on the presence or omission of $M_3$ and/or $N_5$ and it is possible to visualize the subgraph that lead to this classification by leveraging the `visualize_graph` function. By doing this, we retrieve the following visualizations:

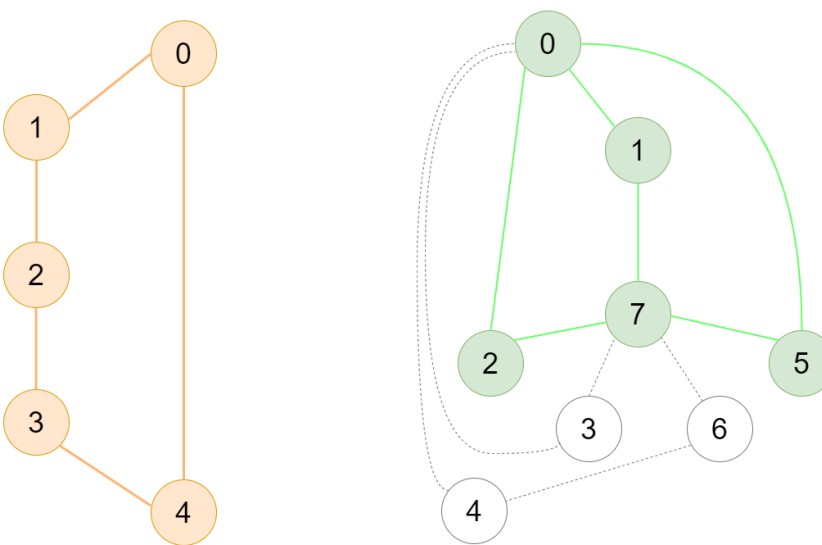

Figure 10: Visualizations obtained with GNNExplainer on weak distributive generalization (on the left) and strong multiclass generalization (on the right)

On the right, the substructure identified as $N5$ by `GNNExplainer` which lead to the classification of said graph as non modular and non distributive. On the right, in green $M3$. Our hierarchical model arrives to the same conclusions as the standard explainer but can also be augmented with a standard explainer.

# H   Code, Licences, Resources

**Libraries**   For our experiments, we implemented all baselines and methods in Python 3.7 and relied upon open-source libraries such as PyTorch 1.11 (33) (BSD license) and Scikit-learn (34) (BSD license). To produce the plots seen in this paper, we made use of Matplotlib 3.5 (BSD license). We will release all of the code required to recreate our experiments in an MIT-licensed public repository.

**Resources**   All of our experiments were run on a private machine with 8 Intel(R) Xeon(R) Gold 5218 CPUs (2.30GHz), 64GB of RAM, and 2 Quadro RTX 8000 Nvidia GPUs. We estimate that approximately 100-GPU hours were required to complete all of our experiments.