# OpenReview forum: "Interpretable Graph Networks Formulate Universal Algebra Conjectures"
_NeurIPS.cc/2023/Conference — NeurIPS 2023 poster_

### Official Review · Reviewer_swnT · 2023-06-27

**Soundness:** 3 good
**Presentation:** 3 good
**Contribution:** 3 good
**Rating:** 6
**Confidence:** 3

**Summary:**

This paper introduces the novel application of Artificial Intelligence (AI) in the field of Universal Algebra (UA), which has remained unexplored until now. The authors propose a methodology that combines equational and topological characterizations with graph neural networks to investigate UA conjectures. To address the limitations of transparency and explainability in graph neural networks, they develop an algorithm for generating AI-ready datasets based on UA conjectures and introduce a new interpretable graph network layer. Experimental results demonstrate that interpretable graph networks offer enhanced interpretability without compromising task accuracy, successfully validate existing conjectures, and identify subgraphs that suggest the formulation of new conjectures. Overall, this paper presents a pioneering approach that integrates AI with UA, enabling empirical validation and formulation of conjectures in this foundational field of mathematics.

**Strengths:**

This paper propose to use GNN to study the UA subject in the modern mathematics.

The paper claims to have generated the first UA dataset, which is a previously untouched area.

This paper also proposed the iGNN which improves model interperability and identifies the relevant subgraphs.

**Weaknesses:**

As the authors pointed out, the method only applies to finite number of lattices.

**Questions:**

na

**Limitations:**

Limited to finite lattices: The methodology is restricted to finite lattices, which may not capture all the relevant information about infinite algebraic structures. While insights gained from finite-lattice explanations can still provide valuable information, the generalization to infinite structures may be limited.

Restriction to topological properties: The approach focuses on topological properties on graphs, and it may not be applicable to non-structural properties that require the use of other interpretable models. While the method is effective in characterizing equational and topological conjectures, it may not cover a broader range of mathematical properties that are not solely determined by the graph structure.

---

> ### Author Rebuttal · Authors · 2023-08-09
>
> We express our sincere appreciation to the reviewer for demonstrating a comprehensive grasp of the fundamental aspects of our paper and for the positive comments about the importance of the problem we investigate, as well as the quality of the presentation and the experimental evaluation.
>
> *“As the authors pointed out, the method only applies to finite number of lattices.”*
> The methodology at hand is restricted to the analysis of a finite number of lattices, which is an inevitable consequence of the finite nature of datasets. However, the conjectures derived through the application of our methodology have the potential to encompass infinitely many lattices (as well as infinitely-sized lattices). This is evidenced by the experiments illustrating the Dedekind and Birkhoff theorems 2.3 and 2.4 in the paper, which are known to hold for the entirety of the modular and distributive lattice classes, respectively. In addition, we observe that known and relevant lattice omissions often rely on lattices with few nodes [8,9,10], hence dealing with (small) finite lattices does not seem to significantly affect our approach.

---

### Official Review · Reviewer_3uvy · 2023-07-06

**Soundness:** 3 good
**Presentation:** 2 fair
**Contribution:** 3 good
**Rating:** 4
**Confidence:** 3

**Summary:**

In this paper, the authors explored the use of AI in mathematical problems Universal Algebra. They propose an algorithm for generating datasets of lattice varieties and derive an interpretable graph neural networks for predicting universal algebra’s properties. Experiment results show the effectiveness of the proposed method. Overall, the motivation is clear and the paper is well written.

**Strengths:**

The problem the paper is investigating is of interest and importance. The authors propose a relatively complete pipeline consisting of both datasets generation and a graph neural network for interpretable prediction.

The paper is well written and the problem definition is clear. In addition, the authors provide examples to emphasize how the Gumbel softmax works which makes it more readable.

Experiment results seem reasonable. It is good that the authors verify their method on datasets with different settings and use qualitative metrics for interprebility.


**Weaknesses:**

One concern is from the dataset generation. As the authors mentioned that the number of existing lattices increases exponentially as n increases, I am curious whether the algorithm is really scalable. In addition, it seems that the algorithm is unable to control label distribution generated.

Using Gumbel softmax instead of regular activation functions seems novel. However, one concern is whether the semantic concept acquired via the one-hot embedding is really meaningful. Because two embeddings may not necessarily have correlation when they have the maximum value in the same indice.

The authors mainly compare their method with black-box GNNs and show better results. However, no recent works are included for comparison. Though the authors discussed the difference between the proposed method with existing general graph explanation methods. There are actually some ad-hoc explanation methods that do not rely on subgraph patterns, e.g. GSAT.

Ref:
Miao S, Liu M, Li P. Interpretable and generalizable graph learning via stochastic attention


**Questions:**

What is the running time for generating the datasets used in the experiments? How is the time increased with n?

Can the authors include recent general graph explanation methods for comparison?


**Limitations:**

It is recommended that the authors can provide results in a table to be more readable.

---

> ### Author Rebuttal · Authors · 2023-08-10
>
> We thank the Reviewer for the positive comments.
>
> *“Concerning the number of lattices and algorithm scalability”*
> The scalability of the dataset generation process is related to two main points: (i) dealing with the exponential growth of the number of lattices, (ii) how we may feasibly generate a lattice on a finite set of elements.
> (i) Since the number of lattices definable on a set L={0,...n-1} increases exponentially with n, it is not feasible to realize a dataset containing all lattices of n elements. However, this is also unnecessary for the scope of this work, which aims at automatically identifying (small) topological patterns responsible for the failure of certain algebraic properties. In addition, the majority of the graphs generated are isomorphic, thus not particularly informative for our task. Because of this, we only generate a portion of all (see point (ii)) lattices up to a certain value $M$, and then take a fixed number of samples $n_s$ up to $N$ ($M=8, n_s = 20, N=50$ in the paper).
> (ii) Given a set of nodes L={0,...,n-1}, each “candidate” lattice is stored as a $n\times n$ matrix with a 1 in each position $(i,j)$ meaning that  $i\leq Lj$, and 0 otherwise, where $\leq_L$ represents a binary relation on $L$.Then we constraint each matrix to have 1 in the diagonal (for reflexivity), and 0 in each position $(i,j)$ with $j<i$, where $<$ is the order of integers (this allows for the strong pruning of isomorphic lattices per cardinality, see (i), without losing any lattice configuration and while ensuring anti-symmetry). All the remaining pairs of elements $(i,j)$ can be either such that  $i\leq_L j$ or incomparable (i.e. $i \nleq_L j$ and $j\nleq_L i$). Since, the obtained matrix not always represent a transitive relation, we apply (truncated) matrix multiplications of each matrix with itself, to get the transitive closure of the order relation (this step converges in at most $n-2$ iterations), hence, assuring that the relation $L$ represents a partial order.
> Following Theorem A.5, we then check if each pair of elements $(i,j)$ admits a (unique!) infimum and supremum (check lattice property) - please refer to lattice.py in the supplementary material for implementation details. The lattice checking and all the lattice operations are translated to standard optimized tensors operations to leverage GPU quicker computations, which becomes particularly advantageous when dealing with lattices’ dimensions for which the computational cost of Algorithm 1 surpasses the overhead of GPU communication. For more details on the complexity of these operations please see the answer to question #1.
> Finally, the complexity of checking an equational property on a lattice depends on the number of variables and on the lattice terms involved in the equation, e.g. is $n^3$ for the distributive property.
> We will include these considerations in the Appendix.
>
> *“Control label distribution generated.”*
> In our experiments we control the label distribution of train and test splits using sub-sampling. However, this process can be implemented directly in the lattice generation algorithm by filtering the lattices that fulfill the desired properties after applying the "isLattice" function within Algorithm 1.
>
> *“Using Gumbel softmax (...) seems novel (...) whether the semantic concept acquired via the one-hot embedding is really meaningful. ”*
> In our setting, the nodes’ features are ignored, hence the message-passing algorithm guarantees that similar embedding representations h_i will correspond to nodes with the same neighborhood. Nevertheless, the concepts we want to identify correspond to subgraphs whose nodes’ neighborhood may differ. However, even in this case nodes activating the same concept encoding all share a common pattern i.e., the max value of the node embedding. As an example, we can simply consider the differences in the neighborhood between the top and an intermediate node in an $M_3$ lattice, even if they participate in the same concept.
>
> *"Missing comparisons, e.g. GSAT”*
> The  submitted paper already compares iGNNs with GNNExplainer (Fig 10, examples of subgraph explanations) and GCExplainer (Fig 4 for fidelity and completeness, note that: GCExplainer should replace the label “Black-box GNN” in the legend)In the revised version we also include a quantitative comparison with GSAT (see tables in the PDF attached to the main rebuttal), and  observe that GSAT and iGNNs obtain comparable results in terms of task generalization and concept completeness when trained on the proposed UA’s tasks. We thank the reviewer for bringing this method to our attention as it allowed us to strengthen the experimental results.
>
> *“What is the running time (...)?”*
> The running time increases polynomially in the size n of the given lattice (it is O(n3) for checking each of the equational properties, e.g. distributivity/modularity, and O(n4) to check if a candidate binary relation is actually a lattice). In a machine with a single quad-core CPU, it requires 20 minutes to generate all the lattices up to n = 8 and sampling 20 lattices for n  [9, 50] (the dataset we used in the paper).
> We will integrate the details on the running time in a new section of the appendix and the complexity of the algorithm in its description in Section 3.1.
>
> *“Can the authors include (...) comparison?”*
> See the uploaded new tables in the common rebuttal answer.
>
> *“It is recommended (...) provide results in a table…”*
> Some details of the main key findings were already summarized in the tables 1 and 4 present in appendices C, and D. We will update these two tables and Figure 4 of the main text including the comparison with GSAT.

---

> > ### Comment · Reviewer_3uvy · 2023-08-21
> >
> > Thanks for the authors' effort on discussing my questions. Most of my questions have been addressed. However, from the comparison between iGNN and GSAT, they achieve comparable performance and GSAT is also ad-hoc mechanism. The novelty of the proposed method is undermined. I would like to keep my current ratings.

---

> > > ### Author Response · Authors · 2023-08-21
> > >
> > > “*The novelty of the proposed method is undermined*”.
> > >
> > > First, we would like to highlight that our novel architecture is one of the three contributions presented in the paper. In any case, we do not agree with the Reviewer for 3 reasons:
> > >
> > > 1- **Novelty in results**: our results show that iGNNs do attain better results than GSAT on one of the key metrics: *The completeness score of iGNNs in strong generalization mode is better than GSAT by up to 30%*. This shows that the interpretable graph representations learnt by GSAT do not generalize to larger graphs, thus undermining GSAT interpretability while *strengthening our contribution*.
> > >
> > > 2- **Novelty in method**: iGNNs differ from GSAT from a methodological perspective.
> > >
> > > 3- **Novelty in application**: to the authors knowledge, our paper is the first one to apply AI methods to investigate lattice equational properties through graphs and, most importantly, to propose a graph-based neural model capable of automatically identifying known conjectures in universal algebra.
> > >
> > > Furthermore, we remark that iGNNs are not ad-hoc models for universal algebra^, as they can be applied on top of (almost) any GNN to make it interpretable through the identification of relevant sub-patterns in a graph classification task.
> > >
> > >  ^if we interpret the reviewer's concern “*GSAT is also ad-hoc mechanism*” correctly.

---

### Official Review · Reviewer_LfMh · 2023-07-07

**Soundness:** 3 good
**Presentation:** 2 fair
**Contribution:** 2 fair
**Rating:** 5
**Confidence:** 2

**Summary:**

This paper aims to use AI as a tool to help solve problems in mathematics. In particular, this paper focuses on universal algebra, which studies algebraic structures and can be considered as learning properties of graphs.

The contributions are two folds:
- This work proposes a method for generating a benchmark for universal algebra, and releases a dataset of 29000 lattices (partial orderings, which can be represented as graphs) of different sizes and 5 properties.
  - Generation: filter binary matrices (each lattice on a set of size n can be represented by a nxn matrix) that admits a partial ordering.
- This work proposes an interpretable neural layer (iGNN) that can be used to solve tasks in the benchmark dataset.

Empirical results show that the model is able to generalize across lattice sizes while being interpretable.

**Strengths:**

- The paper provides the required background and is self-contained.
- The proposed benchmark dataset is the first dataset for universal algebra which is valuable to the field.
- The proposed interpretable layer achieves good empirical performance.
  - Experiments in Section 5.1 show that on UA tasks, the proposed iGNN scores higher in terms of interpretability metrics, while preserving the performance of black-box GNNs.
  - The interpretability results in Section 5.2 are interesting: the appearance of substructures corresponding to N5 and M3 is consistent with the theoretical understanding.

**Weaknesses:**

- The dataset generation process is relatively straightforward (i.e. filtering to get valid lattices, and sampling since it's computationally infeasible to enumerate all configurations).
  - Update post-rebuttal: this is no longer a concern: the author responses have discussed more details and considerations on scalability.
- The proposed neural layer is interpretable due to constraining the function classes, which may limit the representational power of the network.
  - For the pooling layer, the paper chooses max or sum pooling, which reflects the existence or number of 1s in the pooled inputs; this seems to provide only a limited level of interpretability and I wonder if other aspects can be used.

**Questions:**

- Sec 3.2.1, node level features: the "vice versa" part is unclear: we only have the converse direction if we assume that the mapping has some reverse Lipschitz condition (e.g. if the neighborhoods are distinct, then the corresponding $h_i$'s are different), which is not guaranteed in the current setup?
- top of page 5: where do the values of $h_{II}$ come from? Are the values made up as examples?
- Sec 3.2.3: what are examples of intermediate loss functions? How to obtain the signals/labels for the loss?
- Sec 5.2: what does a "concept" mean? How are the weights computed? Is there a set of "concepts" (small graphs?) to search over?

Writing related:
- A notational question: why $\approx$ rather than =?
- The background can be defined more clearly in the main text (or please consider adding pointers to the appendix):
    - explain that $\vee, \wedge$ correspond respectively to join (least upper bound) and meet (greatest lower bound).
    - explain $M_3$ and $N_5$ are named for "Modular" and "Nonmodular".
    - line 77: is there a typo?
    - to help understand the properties, it might be helpful to point out that not modular implies not distributive.
- Sec 3.1: are $n,L$ the same thing?
- Fig 6: where's $M_3$?

**Limitations:**

The paper discusses limitations that the proposed methods are constrained to finite-size graph and only work with structural properties.

There is no direct societal impact.

---

> ### Author Rebuttal · Authors · 2023-08-10
>
> We thank the reviewer for the constructive feedback on the paper. In the following paragraphs we address the raised comments, in the hope of shedding light on the significance of our research. .
>
> *“The dataset generation process is relatively straightforward.”*
> We disagree on considering a straightforward data generation process as a weakness. This dataset is the first benchmark for studying UA lattices with AI, thus constituting a scientific contribution by definition. It follows a detailed explanation of why we think the data generation process is also not as straightforward as it may seem at first sight.
> First, notice that a brute-force approach is infeasible for large lattices, as given a set of n nodes, the number of binary relations on this set is $2^{n^2}$.  To cope with this issue, first for each candidate lattice **L** we consider a squared $n\times n$ matrix representing its order $\leq_L$ having a 1 in each position $(i,j)$ if and only if the element $i$ is less or equal to the element $j$ in **L** (i.e. $i\leq_L j$) and 0 otherwise. Then we constraint each matrix to have 1 in the diagonal (reflexivity), 0 in each $(i,j)$ with $j<i$ , where $<$ denotes the order on $\mathbb{N}$ (this both prunes the majority of isomorphic lattices and yields anti-symmetricity). All the other pairs of elements $(i,j)$ can either be s.t.  $i\leq_L j$ or incomparable (i.e. $i\nleq_L j$ and $j\nleq_L i$), and we consider all these possible cases. Finally, we apply matrix multiplications to get a transitive closure of the order relation (converge guaranteed in at most n-2 steps) and  $\leq_L$ represents a partial order.
> To assure the order represents a lattice, we have to check that each pair of elements $(i,j)$ admits a (unique!) inf and sup. This step and checking lattice equational properties are implemented tensorially to leverage GPU quicker computations, hence being particularly advantageous when the dimensions of the lattices is s.t. the computational cost of Alg 1 surpasses the overhead of GPU communication.
> Lastly, even removing all isomorphic lattices, for $n=18$ there are around 165Bn non-isomorphic different lattices [2]. This is why we sampled a fixed number of lattices as $n$ increases, e.g. 20 samples for cardinality after a certain threshold, instead of keeping generating all the possible lattices for each value of $n$, which is not particularly relevant for our task. Whereas it allows us to study the strong generalization capability of GNNs trained on e.g. up to $n=8$ nodes and then evaluated on lattices of higher dimensions, e.g. $n=50$ nodes (see Fig. 5 of the main paper). Notice that checking e.g. distributivity for $n=50$ is deterministic but requires verifying $50^3$ identities checking.
> For completeness we will include these considerations in the Appendix.
>
> *“The proposed neural layer is interpretable due to constraining the function classes, which may limit the representational power of the network.”*
> Interpretable models constrain the representation power by definition (see Rudin [3], or CBMs [4] vs DNNs, or Decision Trees vs XGBoost). The problem therefore is how much these constraints affect the classification accuracy. Our results empirically demonstrate that these constraints do not deteriorate the task accuracy (see Figs 4-5).
>
> *“Sec 3.2.1, node level features: the "vice versa" part is unclear (...)”*
> We notice that for the UA’s properties investigated, only the pure graph structure is relevant while (input) node features do not play any role. In this context, relational models may not consider node features at all (e.g., KGEs) or generate constant input features to allow message passing. Hence, we get similar embeddings $h_i\approx h_j$ after message passing if and only if i and j have the same neighbors. We clarified it as mentioned in the list of changes.
>
> *“top of page 5: are h_II values made up as examples?”*
> Yes.
>
> *“Sec 3.2.3: what are examples of intermediate loss functions?"*
> The intermediate supervisions on HiGNNs mentioned in Sec 3.2.3 are applied on the task predictions $\hat{y}^{(l)}$ and obtained via Eq. 5 at each layer for $l=1,...,K$ (where $\hat{y}^{(K)} = \hat{y}$). The task labels (i.e., $y$) and the loss function (e.g., cross-entropy in our experiments) are always the same and can be used to supervise any layer.
>
> *“Sec 5.2: what does a "concept" mean?”
> The notion of “concept” in general follows Ghorbani et al. [5] (lines 146-7), while in the context of GNNs, we follow Magister et al. [6], where a concept is represented by a subgraph.
>
> *“A notational question: why \approx rather than =?”*
> “$\approx$” distinguishes from “$=$” as it denotes equations on variables instead of elements.
>
> *“background can be defined more clearly in the main text”*
> We revised the background section addressing all the suggestions.
>
> *“Sec 3.1: are n, L the same thing?”*
> L is the set of elements of a lattice, while $n=|L|$. We replaced the typo $L=10$ to $n=10$ in Sec 3.1.
>
> *“Fig 6: where's M_3?”*
> In Fig 6, M3 is a sublattice of lattices number 2,4,5 (left) and of 2,3,5 (right). We will add the prefix “(sub-)” in front of “lattice” in the caption of Fig 6.

---

> > ### Comment · Reviewer_LfMh · 2023-08-20
> >
> > Apologies for my late reply! Thank you very much for the clarifications; I have raised my score.
> >
> > May I have some further clarifications please:
> >
> > - About limiting the representation power: the current experiments do show that the representation power of iGNN is sufficient for solving these UA tasks, but I was wondering whether/how iGNN would be limiting in general: could you comment on what tasks would black-box GNNs be able to solve but iGNN would not? How is this related to the observation that black-box GNN outperforms iGNN on smaller number of nodes?
> > - Clarification about strong vs weak generalization: is the "strong vs weak" distinction defined somewhere?

---

> > > ### Author Response · Authors · 2023-08-21
> > >
> > > Thank you for raising your score and engaging the discussion! We know it’s not the ideal time of the year to serve as a reviewer.
> > >
> > > Here we provide further clarifications on the raised questions.
> > >
> > > **About limitation of the representation power**
> > > The representation power of iGNNs is generally constrained by the size of the concept embeddings $q$, unlike standard GNNs. In practice, the Boolean nature of concept embeddings limits the range of representable concepts to $2^q$, rendering iGNNs unable to perfectly handle tasks requiring more than $2^q$ concepts. Additionally, the Gumbel Softmax's enforcement of Boolean representation leads to more rigid and bumpy training, especially when working with datasets containing limited informative samples, as seen in small graphs in our experiments.
> > >
> > > Also in our experiments we used an interpretable classifier based on a linear layer because it was enough to solve the tasks. However, for more challenging tasks (e.g.,dependent on a non-linear combination of concepts) other interpretable and differentiable models could be considered, without the need to change the backbone structure of iGNNs.
> > >
> > > **Strong vs weak generalization definition**
> > > We draw the distinction between "strong vs weak" generalization based on [1], which uses the term "strong generalization" to describe GNN models' performance on larger graphs than their training set. This type of experiment falls under out-of-distribution testing, a highly challenging scenario for statistical learning models.
> > > We will add this reference to the main paper when introducing the two tasks.
> > >
> > > [1] Veličković, Petar, et al. "Neural execution of graph algorithms." arXiv preprint arXiv:1910.10593 (2019).

---

> > > > ### Comment · Reviewer_LfMh · 2023-08-21
> > > >
> > > > (I guess there is never an "ideal" time to serve as a reviewer, so it really is my fault rather than the timing, but thank you for your understanding :))
> > > >
> > > > Thank you for these further clarifications!

---

### Official Review · Reviewer_AeQ9 · 2023-07-07

**Soundness:** 3 good
**Presentation:** 3 good
**Contribution:** 2 fair
**Rating:** 6
**Confidence:** 3

**Summary:**

A general methodology is introduced that allows a researcher interested in problems regarding lattices, in the mathematical field of Universal Algebra, to investigate any algebraic property whose validity can be verified on a finite lattice.
The paper indicates how a dataset can automatically be generated . Then a new GNN variant is devised, adapted from previous works (e.g. [1]), that is interpretable. This model is compared to the performance of a vanilla, black-box GNN, with all the other model parameters kept identical.

[1] Steve Azzolin, Antonio Longa, Pietro Barbiero, Pietro Liò, and Andrea Passerini. Global ex-
plainability of gnns via logic combination of learned concepts

**Strengths:**

The paper discusses how a field that has hitherto not seen a large number of AI applications be treated. Two new GNN variants are proposed, called iGNN and HiGNN.
I also like the fact that the well-known lattices N_5 and M_3 where used throughout the text as examples of how the learning algorithms works (Example 3.1 and 3.2).

**Weaknesses:**

__I. Various hyperbolizing adjectives are used throughout the text, that do not provide information:__

- line 18: "Universal Algebra (UA, (6)) is one of the foundational fields of modern mathematics" The importance of Universal Algebra is somewhat overstated. While it indeed is a foundational field, it is usually not part of a typical university curriculum.

- line 43 onwards: "First, we propose a novel algorithm that generates a dataset suitable for training AI models based on an UA equational conjecture. Second, we generate and release the first-ever universal algebra’s dataset compatible with AI [...] Our findings
demonstrate the potential of our methodology and open the doors of universal algebra to AI."
The relevance of the contribution is hyperbolized and overstated, for the following reasons:
1) Indeed, this may be the first application of AI to universal algebra (although experience has shown that in machine learning often new results are introduced, that under a different name were already known previously -- --, so unless an extremely thorough literature review would have been made, I would not feel confident making this assertion).
2) It is known that AI is very well suitable to a particular type of mathematics, where the involved objects are discrete (see [1]), so it is not surprising that this approach works well in this setting too.
3) I am not convinced paper does not "open doors of universal algebra to AI" (see also the claim repeated on line 408), since the method is not tied to the field of Universal Algebra, but rather to the objects of study. Universal Algebra has strong connections to model theory, for example (see [2], chapter 5, which is a reference that is cited in the paper), and exploring these using machine-learning is definitely not covered under the current approach.

line 90: "advanced AI methods" (what exactly makes an AI method "advanced" is debatable)

line 396: "However, as universal algebra is a foundational branch of modern mathematics, any contribution to this field can already have significant implications in various mathematical disciplines."

I feel like these hyped-up claims obscure the actual strong points of the paper: A well-executed experiment with a suitable neural model (GNN) (and other strong points outlined in the Strength section).

__II. Other observations__

- while not a weakness, but a curiosity: it is noteworthy that the lattice examples from Figure 2, page 2, are identical to those from the Wikipedia article on lattices, see Pic 10 and Pic 10 [3]. In Wikipedia, it is mentioned that these are the smallest lattices of their type, but the paper does not mention this (tangential) fact.

- in the appendix, a number of definitions from Universal Algebra are taken from well-knowns, e.g., Definition A.3 corresponds ad literam to Definition 9.3 from [2]. I feel it could be emphasized more strongly that many of the definitions are directly copied from existing sources, rather than merely stating " More formally (6; 8; 16):", where their reference 6 is my reference [2] below.
More precisely, at the beginning of the appendix, it is stated: "An n-ary operation on A is a function that takes n elements of A and returns a single element from the set. More formally (6; 8; 16):" which implies that only the first ensuing definition of arity is taken from references rather than all ensuing definitions.
It would be helpful, to either explicitly state that all theorems/definitions are taken from books (and ideally, like in Theorem A.5, show from where the theorem comes).

- on table from the appendix is missing (below line 582): "Note: ADD TABLE WITH CONCEPT PURITY". This should be fixed, if the paper is accepted, but doesn't make a difference for acceptance/rejection decision.

[1] A.Z. Wagner, Constructions in combinatorics via neural networks, https://arxiv.org/abs/2104.14516
[2] Burris S., Sankappanavar H.P., A Course in Universal Algebra, Springer Millenium Edition
[3] https://en.wikipedia.org/wiki/Lattice_(order)

**Questions:**

- Is there a reason you chose to use linear classifiers instead of decision trees, as the interpretable model of choice?

- consider line 150 ("To address this, we use a hard Gumbel-Softmax activation") vs. line 219 ("The choice of the activation and loss functions to train iGNNs depends on the nature of the task at hand and does not affect their interpretability"). Does that mean that you use different activation functions at different parts of the GNN?

**Limitations:**

The problem that is solved is quite narrow, though the authors go out of their way to make sure that this is not emphasized.

---

> ### Author Rebuttal · Authors · 2023-08-09
>
> We express our appreciation to the Reviewer for demonstrating a comprehensive grasp of the fundamental aspects of our paper. We comment on the points of the review in the following text.
>
> *“I. Various hyperbolizing adjectives are used throughout the text, that do not provide information: /line 18/line 43/The problem that is solved is quite narrow”*
>
> We thank the Reviewer for the meaningful suggestions. We agree with the reviewer and in the revised version of the paper, we will rephrase the mentioned statements in a more moderate fashion, while contextualizing the novelties introduced by our paper with respect to the specific contributions within the field of UA.
>
> We acknowledge that Universal Algebra (UA) and abstract algebra in general are usually not covered in standard university programs as they constitute topics requiring a strong comprehension of deep mathematical concepts in a broad view. However, we notice that this can be similarly observed for other fields in Mathematics, like Category Theory and axiomatic Set Theory, which are  widely recognized as foundational branches of Mathematics.
>
> We also would like to note that, to the best of our knowledge, UA problems remain largely unexplored by AI methodologies for two main reasons. Firstly, UA primarily deals with infinite objects or even classes of abstract objects, which pose unique challenges for conventional AI techniques. Secondly, the field commonly relies on deterministic algorithms, utilized to construct finite models or serve as deterministic theorem provers, such as "mace4" and "prover9" [1]. In this sense, we hope that our paper could represent a guidance to further explore the study of UA’s problems with AI.
>
> *“while not a weakness, but a curiosity: (...)”*
> We thank the Reviewer for the remark. We will clarify this point in the revised paper when referring to Figure 2 as follows: “Figure 2 represents M3 and N5 that are the smallest instances of lattices which exhibit failures of distributivity and modularity, respectively”.
> Notably, it is not surprising that the depiction of these lattices in our paper aligns with their representation found in Wikipedia since it is the usual way to represent them, following the lattice order.
>
> *“in the appendix, a number of definitions from Universal Algebra (...)”*
> In the revised version, we will make explicit mention which definitions are directly drawn from [11]. Nevertheless, we deemed it necessary to include them in our paper to ensure self-containedness, particularly for an AI audience that may be unfamiliar with the domain of UA.
>
> *“on table from the appendix is missing (...)”*
> The main pdf contains a preliminary version of the appendix, while the finalized version of the appendix can be found in the supplementary material of the submission.
>
> *“Is there a reason you chose to use linear classifiers instead of decision trees (...)” ?*
> Decision trees represent a suitable choice as interpretable models and we have also used them through our investigation, e.g. in the evaluation of the completeness score of non-interpretable models in Figure 4. In devising our approach, we preferred to rely on linear classifiers as they are fully differentiable, hence allowing us to realize a fully interpretable and differentiable model from the input to the classification head. However, in practice any interpretable and differentiable classifier could be used in place of this linear layer.
>
> *“consider line 150 ("To address this, we use a hard Gumbel-Softmax activation") vs. line 219 ("The choice of the activation and loss functions to train iGNNs depends on the nature of the task at hand and does not affect their interpretability"). Does that mean that you use different activation functions at different parts of the GNN?”*
> Line 150 refers to the activation function generating concept encodings (always a Gumbel softmax in our approach, invariant w.r.t. the task at hand), while Line 219 refers to the activation and loss function of the classification layer (usually softmax/sigmoid + cross entropy/MSE, depending on the task at hand).

---

> > ### Comment · Reviewer_AeQ9 · 2023-08-18
> >
> > I thank the authors for their response, in particular the elucidations to the last two questions (I could recommend adding them to the main paper).
> >
> > > Secondly, the field commonly relies on deterministic algorithms, utilized to construct finite models or serve as deterministic theorem provers, such as "mace4" and "prover9" [1]. In this sense, we hope that our paper could represent a guidance to further explore the study of UA’s problems with AI.
> >
> > I'm intrigued by this remark. Could you add more motivation how your method might interact with (or even improve) these theorem provers?
> > i am not too familiar with them, so maybe I am mistaken when I am saying this, but it seems to me that these provers use finite models to create counterexamples to claims?

---

> > > ### Author Response · Authors · 2023-08-20
> > >
> > > We extend our gratitude to the Reviewer for the appreciation of our response and for engendering a thought-provoking discussion.
> > >
> > > “*the elucidations to the last two questions (I could recommend adding them to the main paper).*”
> > >
> > > In accordance with the Reviewer's suggestion, we intend to add two additional sentences within the revised version of the paper, clarifying the central aspects of the two questions.
> > >
> > > “*I'm intrigued by this remark. Could you add more motivation how your method might interact with (or even improve) these theorem provers? i am not too familiar with them, so maybe I am mistaken when I am saying this, but it seems to me that these provers use finite models to create counterexamples to claims?*”
> > >
> > > “*mace4*” is a “model-maker” that is used mainly to investigate finite models with a certain axiomatisation. The Reviewer correctly identified the main utility of this tool in searching counterexamples, however it is also used in various other tasks. For instance, it can be useful also to improve the comprehension of the structures that model specific axioms and to understand how they change removing some of the axioms or adding others (i.e. which of them are not satisfying certain additional axioms?, removing one of the axioms which models are included?, and so on).
> > >
> > > “*prover9*” is instead a classic theorem-prover that, given a set of axioms and a statement to prove, outputs a proof of the claim if it is able to find it.
> > >
> > > Both of these tools are quite known in algebra and they can be integrated by our framework which provides an orthogonal information w.r.t to the ones given by “*prover9-mace4*”. Indeed, the process of discovery in mathematical research involves a loop of the following steps:
> > >
> > > 1- Having an idea about a mathematical property.
> > >
> > > 2- Conjecturing a statement.
> > >
> > > 3- Either prove the statement or find a counterexample.
> > >
> > > 4- Refine the idea and the conjecture, if not proved.
> > >
> > > Clearly, “*prover9-mace4*” are mainly used for step 3 of this loop, while our framework aims to help phase 1-2 (usually the most difficult parts) providing a better understanding of structures and eventually leading to new ideas.
> > >
> > > To illustrate, let us suppose for example that for some reason we are interested in introducing and studying new mathematical structures characterized by a set of axioms A (a common problem in algebra, logic, and other fields). Then, *mace4* can be used to conduct a preliminary investigation, to understand how the finite models of A look like for small cardinalities usually (in the case of Algebra the first questions are: How many finite models of small cardinality there are? Which are the congruences of these structures?, Can they be characterized in a specific way?, Which are the subdirectly irreducible structures?, and so on).
> > >
> > > Then, our framework could be applied to the congruence lattices of those structures, which often encode many relevant properties of the structures, looking for recurrent patterns that can characterize these properties (as a new conjecture).
> > >
> > > In the last part *prover9* and *mace4* could be then used to prove or disprove the conjectured statement.
> > >
> > > The authors once again express profound gratitude for the keen interest demonstrated by the Reviewer and for the highly stimulating discourse that has emerged from her/his insightful observations.

---

> > > > ### Comment · Reviewer_AeQ9 · 2023-08-21
> > > >
> > > > Unfortunately, there is not enough time now; but showing how your work could be integrated in a loop with prover9 and mace4 seems to me to really be the biggest original contribution (otherwise it is just a technical improvement over the current state of the art); I will leave my score as it is for now, but perhaps you will be able to find time to make this update for the camera ready version. At least including an explanation in the paper, as you did above, would be highly beneficial for readers.

---

> > > > > ### Author Response · Authors · 2023-08-21
> > > > >
> > > > > Thank you once again for your feedback, we will include a summary of this discussion in the camera ready. We do agree that the connection with existing provers could be an interesting way forward of practical use for algebrists.

---

### Author Rebuttal · Authors · 2023-08-09

We thank all the reviewers for their comments and useful suggestions. In the following, we address the main concerns and questions reviewer-by-reviewer. We also describe minor changes we made based on these suggestions in the reviewed version of the paper, however, the core of our work’s contribution remains unchanged. We earnestly hope that our responses to the reviewers' requests have been sufficiently satisfactory. However, we remain committed to address any additional concerns or queries that may arise during the discussion period, with the ultimate aim of further refining and enhancing the quality and impact of our research work.

**List of changes in the revised version of the paper:**

- Comparison with GSAT added in Table 1 and Table 4 of appendix C and in Figure 4 of the main text (see PDF attached to this comment).
- In line 18 and 19  we rephrase the mentioned statements to contextualize the novelties introduced by our paper with respect to the specific contributions within the field of UA.
- At line 152, we replace the sentence “(...) where \psi and \phi are (...)” with the sentence “(...) where \psi is a learnable function ignoring or assuming constant input features and \phi is (...)”.
We change the sentence at line 67  “(...) every pair of elements has a supremum and an infimum (...)” with the sentence “(...) every pair of elements has a supremum (∨) and an infimum (∧) (...)”.
- Line 77 “it” to “the latter”.
- At line 82 when referring to Figure 2 we modify the sentence to “For instance, Figure 2 represents M3 (modular with 3 atoms) and N5 (non-modular with 5 elements), the smallest instances of lattices which exhibit failures of distributivity and modularity, respectively”.
- We will add the sentence: “Furthermore, we notice that each distributive lattice is also modular, see Appendix A.” after the previous one and we will add the half-line proof of this fact in the appendix.
- Line 119  “L=10” to “n=10”.
- We add the prefix “(sub-)” in front of the word “lattice” in the caption of Figure 6.
- We make explicit mention to which definitions of the appendix A are directly drawn from [11].
- Further considerations about Algorithm 1’s complexity, strategy used for the filtering process, and explanation of the tensorial optimization are now added to a new appendix section.

We notice that the main pdf contains a preliminary version of the appendix, while the finalized version of the appendix can be found in the supplementary material of the submission.

Throughout all the rebuttals, we will use the following common reference numbers.

References:
[1] https://www.cs.unm.edu/~mccune/prover9/
[2] Heitzig, Jobst, and Jürgen Reinhold. "Counting finite lattices." Algebra universalis 48.1 (2002): 43-53.
[3] Cynthia Rudin. Stop explaining black box machine learning models for high stakes decisions and use interpretable models instead. Nature machine intelligence, 1(5):206–215, 2019.
[4] Koh, Pang Wei, et al. "Concept bottleneck models." International conference on machine learning. PMLR, 2020.
[5] Amirata Ghorbani, James Wexler, James Y Zou, and Been Kim. Towards automatic concept- based explanations. Advances in Neural Information Processing Systems, 32, 2019.
[6] Lucie Charlotte Magister, Dmitry Kazhdan, Vikash Singh, and Pietro Liò. GCExplainer: Human-in-the-loop concept-based explanations for graph neural networks. arXiv preprint arXiv:2107.11889, 2021.
[7] Heitzig, Jobst, and Jürgen Reinhold. "Counting finite lattices." Algebra universalis 48.1 (2002): 43-53.
[8] Garrett Birkhoff. On the structure of abstract algebras. In Mathematical proceedings of the Cambridge philosophical society, volume 31, pages 433–454. Cambridge University Press, 1935.
[9] Richard Dedekind. Über die von drei Moduln erzeugte Dualgruppe. Math. Ann., 1900.
[10] Bjarni Jónsson and Ivan Rival, Lattice varieties covering the smallest non-modular lattice variety, Pacific J. Math, 1978.
[11] Stanley Burris and H. P. Sankappanavar. A Course in Universal Algebra. Springer, 1981.

---

### Decision · Program_Chairs · 2023-09-21

**Decision:**

Accept (poster)

**Comment:**

### Summary

This paper presents an application of graph neural networks (GNNs) in Universal Algebra (UA), which was an uncharted territory until now. UA serves as the foundation of modern mathematics, and this work pioneers the use of AI to investigate UA conjectures with equational and topological characterisations. While traditional approaches have struggled with these problems for decades, the authors propose a new methodology that leverages graph neural networks for analysis.
The authors introduce an algorithm for generating AI-ready datasets based on UA conjectures to overcome the transparency and explainability issues associated with standard neural networks. Furthermore, they developed an interpretable graph network layer, iGNN, designed to enhance interpretability without sacrificing task accuracy.
The paper's contributions include the generation of a benchmark dataset comprising 29,000 lattices of varying sizes and five different properties, along with the introducing of the interpretable neural layer (iGNN) to address UA-related tasks. Empirical results demonstrate that the proposed method successfully generalises across lattice sizes while maintaining interpretability.
In summary, this paper, AFAIK, marks the first introduction of AI into UA, offering a methodology that integrates equational and topological characterisations with interpretable graph neural networks. It validates existing UA conjectures and suggests the formulation of new conjectures, ultimately paving the way for AI-assisted mathematical exploration in this foundational field of mathematics.

## Decision

I was on the fence for this paper as the reviews were mostly positive but borderline. I recommend this paper acceptance due to:

1. It is addressing and exploring a very interesting problem.
2. The paper is well-written and clear.
3. The authors have done a great job answering and addressing the issues raised by the reviewers.

Thus I think this paper would be of interest for the general NeurIPS community.